# Spurious numerical mixing under strong tidal forcing : a case study in the South East Asian Seas using the Symphonie model (v3.1.2)

Adrien Garinet[1, 2], Marine Herrmann[1], Patrick Marsaleix[1], and Juliette Pénicaud[1]

[1]Université de Toulouse, LEGOS (CNES/CNRS/IRD/UT3), Toulouse, France
[2]Direction Générale de l'Armement, Ministère des Armées, Paris, France

**Correspondence:** Adrien Garinet (adrien.garinet@polytechnique.edu)

**Abstract.** The role of mixing between layers of different density is key to how the ocean works and interacts with other components of the Earth System. Accounting correctly for its effect in numerical simulations is therefore of utmost importance. However, numerical models are still plagued with spurious sources of mixing, originating mostly from the vertical advection schemes in the case of fixed coordinates models. As the number of phenomenon explicitly resolved by models increases, so does the amplitude of resolved vertical motions and the amount of spurious numerical mixing; and regional models are no exception to this. This papers provides a clear illustration of this phenomenon in the context of simulation of the South East Asian Seas, along with a simple way to reduce it. This region is known for its particularly strong internal tides and the fundamental role they play in the dynamic of the region. Using the Symphonie ocean model, simulations including and excluding tides and using a pseudo third-order upwind advection scheme on the vertical are compared to several reference datasets, and the impact on water masses in assessed. The high diffusivity of this advection scheme is demonstrated, along with the importance of accounting for tidal mixing for a correct representation of water masses. Simultaneously, we present an improvement of this advection scheme to make it more suitable for use on the vertical. Simulations with the new formulation are added in the comparison. We conclude that the use of a higher order numerical diffusion operator greatly improves the overall performance of the model.

## 1 Introduction

Diapycnal mixing plays a fundamental role in the ocean, and precise quantification, localisation and understanding of its mechanism are still active topics of research (Oce, 2022). Considering its major role in the heat uptake of the ocean, its correct representation in climate models is of utmost importance. Yet, a major flaws of ocean models is the tendency of numerical methods used to solve the equations on discrete grids to produce numerical errors resulting in an excessive diffusion of tracer fields often referred to as *numerical mixing*. Considering the sensitivity of numerical models outputs to vertical mixing (Bryan, 1987), this spurious phenomenon becomes particularly troublesome when it occurs across isopycnals, since the physical mixing is much lower in this direction than along the isopycnals and numerical mixing can therefore easily dominates. In this context, the discretization of the vertical tracer advection equation in fixed coordinates model - being terrain following or geopotential

coordinates - and the resulting spurious diapycnal fluxes have long been identified as a major source of spurious numerical mixing (Griffies et al., 2000); and it is still considered as a major issue (Klingbeil et al., 2019).

Due to its advective origin, this spurious mixing is strongly linked to the strength of the flow, and is therefore more sensitive in situations of strong vertical displacements. Even if "quasi-eulerian" coordinates (Leclair and Madec, 2011) adjusting vertically with the motion of the free surface are now commonly used in ocean models, being terrain following or geopotential (see e.g. Adcroft and Campin, 2004, for the adaptation to geopotential coordinates), therefore reducing the vertical velocity relative to the mesh, spurious mixing is still troublesome in the current class of global ocean models (Megann, 2018; Holmes et al., 2021). Besides, it might become even worse as the resolution increases and more physical processes are being resolved explicitly - such as submesoscale and the strong vertical speeds anomalies associated with e.g. frontogenesis (Siegelman et al., 2020); or tides and the associated internal waves field, especially in regions of strong internal tide activity. In this sense, regional ocean models are particularly at risk, especially when run over long periods of time.

If the issue is known in the community of model developers, it is unsure whether the fast growing community of model *users* is fully aware of this issue. Since the early 2000's, a number of studies have indeed focused on diagnosing spurious numerical mixing in idealised simulations (see for instance Burchard and Rennau, 2008; Klingbeil et al., 2014; Ilıcak et al., 2012; Gibson et al., 2017). However, the literature on spurious mixing in realistic simulation is still sparse (e.g. Lee et al., 2002; Megann, 2018; Holmes et al., 2021), is focused mainly on relatively coarse resolution global models and relies on rather complex indirect estimates of mixing through water mass transformation analysis. Furthermore, none of the models used included explicit tidal forcing. Yet, owing to the advances in simulating tides in ocean models over the last decades (see e.g. Arbic, 2022) and the role they play in setting the global state of the ocean (Wunsch and Ferrari, 2004), more and more realistic simulations account for tidal forcing and the first few baroclinic modes are now routinely resolved in most regional applications. There is therefore a need to explicitely demonstrate the spurious numerical effect tides can have on tracer fields, even in relatively high resolution models forced at their lateral boundaries. Also, though less precise, comparison against observations - made possible for instance by the growing amount of ARGO profiles in the ocean and availability of high-resolution satellite products - and parameter sensitivity study can help to provide a clear picture of the effect of spurious mixing in realistic simulations.

Now, with the steady increase in available computing power, one can expect the spurious vertical mixing to be eventually reduced to acceptable levels (Holmes et al., 2021), especially since advection schemes are rarely of first order. The number of vertical levels has indeed increased by up to a factor of 10 over the last decades. This however relies on the assumption that the range of vertical speeds resolved by the model does not depend too much on the vertical grid spacing. Furthermore, always increasing the resolution might not be desirable nor achievable in a foreseeable future for every research team and projects around the world, and therefore cannot be regarded as a definitive solution to the issue. Finally, the cost-effectiveness of an increase in resolution depends largely on the order of the schemes used, and both still have to be considered simultaneously (Sanderson, 1998).

Considering the fact that the issue of spurious numerical mixing generated by vertical advection is tightly linked to the (quasi-)eulerian nature of fixed coordinates, moving toward more lagrangian coordinates seems to have been a preferred option in recent years. Such approaches seem promising and have indeed demonstrated capabilities in reducing spurious diapcynal

mixing (Gibson et al., 2017; Megann et al., 2022). In a recent paper, Megann (2024) demonstrated the ability of so called $z-$tilde coordinates (Leclair and Madec, 2011) to effectively reduce spurious numerical mixing in a global eddying simulation forced with tides. However, if they formally cancel the issue of spurious vertical mixing inherent to fixed coordinates models, these approaches come at the cost of a whole range of new issues and limitations that remain a topic of active research, owing to their relative youth (Fox-Kemper et al., 2019; Griffies et al., 2020). Moreover, their implementation into already existing numerical codes requires sustained effort by model developers over years in order to be used in realistic simulations.

There is therefore a need to expand the range of tools available for fixed coordinates ocean models. The goal of this paper is twofold : a new, less diffusive, formulation for advection schemes based on a scheme already implemented in the Symphonie model is proposed; validation of the method through the case study of simulations of the South East Asian Seas is then proposed, providing a clear illustration of both the spurious numerical effect of tides in a regional model and their relevance in the accurate depiction of the region.

## 2 Model configuration

### 2.1 The South-East Asian Seas

#### 2.1.1 Regional context

Many locations have been shown to exhibit intense internal tides activity (Zaron, 2019), correlated with the existence of large areas of strong topographic gradients. Amongst them are the South East Asian (SEA) Seas (see Fig.1), a large region of numerous deep marginal seas and shallow straits connecting the Western Pacific to the Indian Ocean through an on-average westward circulation know as the Indonesian Throughflow (ITF) (Sprintall et al., 2019). Due to this complex bathymetry, the region has long been known as a hotspot of internal tides generation (e.g. Apel et al., 1985), and their dissipation in the various semi-enclosed basins has been recognized as a major driver of the intensified mixing observed there (Ffield and Gordon, 1996). It leads to a sensible transformation of water masses on their way from the Pacific to the Indian Ocean, forming a unique water mass that can be tracked across the basin and beyond, up to the Agulhas Current (Gordon, 2005). This intensified mixing also acts to cool the sea surface temperature (SST) in the region (Susanto and Ray, 2022) and modify the atmospheric deep convection, which in turn impacts the regional and global climate (Koch-Larrouy et al., 2010; Sprintall et al., 2019). Interannual and climatic projections therefore need an accurate modelization of the ocean in the SEA region; and this cannot be achieved without an accurate representation of tidal mixing.

#### 2.1.2 Modelling

In average, the vertical diffusivity over the region is indeed increased by more than one order of magnitude when compared with open-ocean estimates (Ffield and Gordon, 1992), with a high spatial variability (Purwandana et al., 2020); and it has been demonstrated in many studies that models not accounting for tides-induced mixing fail at representing correctly the water masses along the course of the ITF (Koch-Larrouy et al., 2007; Jochum and Potemra, 2008; Nugroho et al., 2018; Sasaki et al.,

2018; Katavouta et al., 2022). Parameterization of tidally driven mixing has been successfully implemented in regional models (see e.g. Koch-Larrouy et al., 2007), but still exhibit some weaknesses (Iskandar et al., 2023). Also, such an approach misses the dynamical effect of tides, which is suspected to play a non-negligible role in the horizontal mixing of water mass properties via the interaction of tidal currents with the complex topography of the region (Hatayama et al., 1996; Nagai et al., 2021), especially for high resolution models. With the increase in computational capacities allowing for simulations to be run on finer grids, thus enabling the explicit resolution of more baroclinic modes, explicit tidal forcing is therefore being used more and more in regional models of the SEA region (Castruccio et al., 2013; Tranchant et al., 2016; Katavouta et al., 2022).

It must be noted however that most of the numerical codes used in those studies solve the primitive equations under hydrostatic assumptions. Considering the resolution of such large regional models, it is likely that this approximation remains valid, insofar as non-hydrostatic phenomenon such as small scales overturning are not resolved. Indeed, differences between hydrostatic and non-hydrostatic models are sensible only when the model horizontal resolution is below a few tens of meters (see e.g. Berntsen et al., 2009; Álvarez et al., 2019); and the good capabilities of such hydrostatic models forced explicitly by tides in reproducing *in situ* observations at a regional scale has been demonstrated in several studies (amongst others, see Tranchant et al., 2016; Katavouta et al., 2022; Thakur et al., 2022; Gonzalez et al., 2023; Bendinger et al., 2023).

Nonetheless, one may wonder how hydrostatic models actually represent tides induced mixing. Formally, intensified mixing is achieved through enhancement of the diffusivity computed either by frictions of tidal currents at the bottom or by the turbulent closure schemes, such as $k - \epsilon$, either through the buoyancy flux or the shear production terms (Gonzalez et al., 2023); and the results have been shown to compare reasonably well with specifically designed internal tide mixing parameterizations (see e.g. Nugroho, 2017; Gonzalez, 2023, chapters 5). However, the question whether this is the fortunate result of several errors compensating each others or if the existing models can actually capture the essential physical aspects of internal waves leading to mixing is still open - especially considering the fact that turbulent closure schemes, although based on first principles, have been designed and calibrated at a time where internal tides were hardly resolved by ocean models. Still, such considerations, though of major interest for the community, are out of scope of the present study, in which we will focus on numerical mixing.

## 2.2 Model configuration and overview

The model employed in this study is the Symphonie ocean model (Damien et al., 2017, and references therein), a three dimensional ocean circulation model solving the primitive equations under Boussinesq and hydrostatic approximations. Symphonie has been used in the past over Southeast Asia at several spatial scales, from the Gulf of Tonkin shelf scale (Piton et al., 2021; Nguyen-Duy et al., 2021) and South Vietnam upwelling coastal scale (To Duy et al., 2022; Herrmann et al., 2023) to the South China Sea regional scale (Trinh et al., 2024). Turbulent closure is achieved through the $k - \epsilon$ scheme (Burchard and Bolding, 2001). Integration in time is peformed using a leapfrog time-stepping algorithm together with a Robert-Asselin filter to damp out spurious numerical modes. Barotropic and baroclinic modes are handled separately following a mode splitting procedure (Blumberg and Mellor, 1987). Momentum is advected via a fourth-order central differencing scheme along with an explicit biharmonic diffusion term, with a viscosity derived from the one developed in Griffies and Hallberg (2000). Since tracer advection schemes are the focus of this paper, the one used in Symphonie are discussed in Sec.2.3.

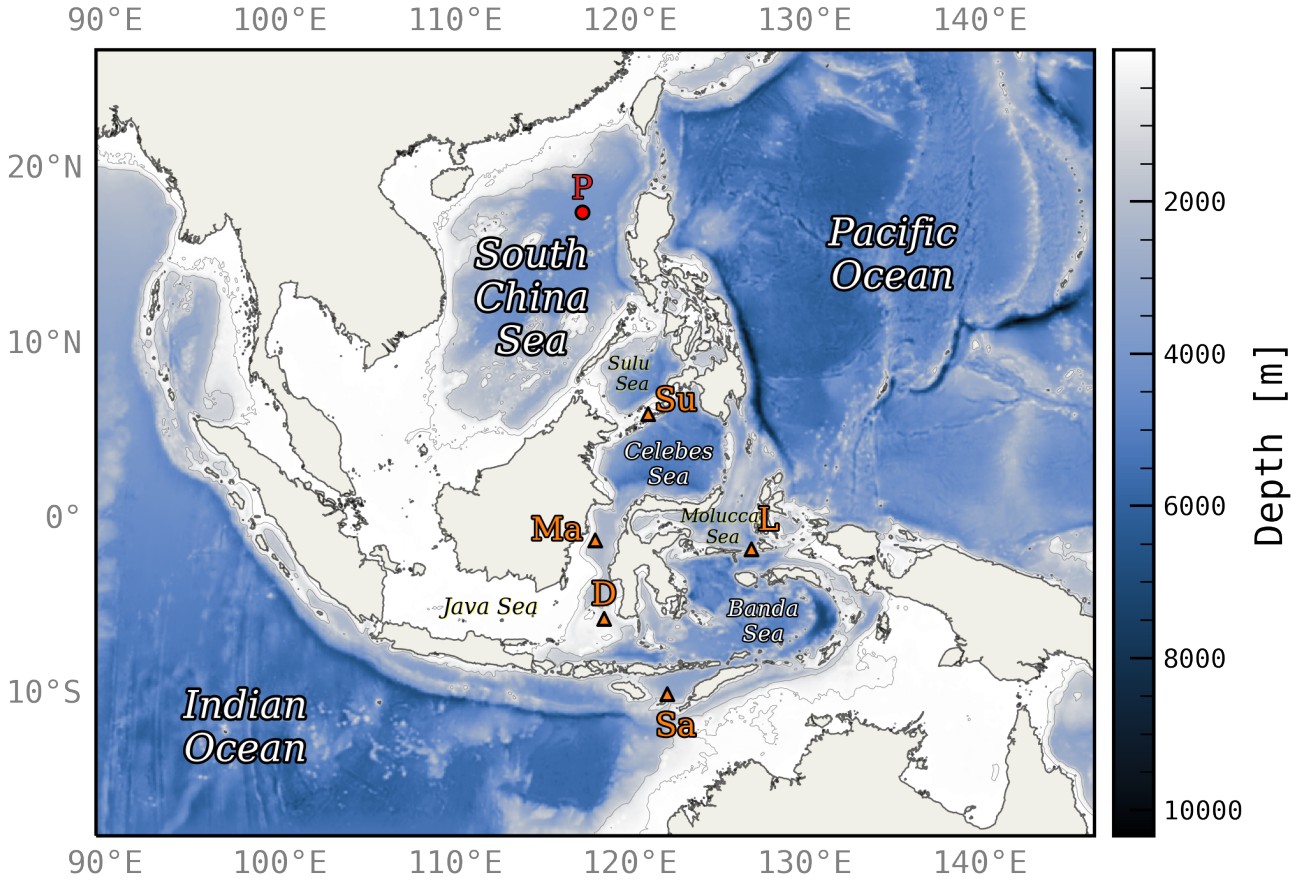

**Figure 1.** Bathymetry of the SEA configuration. The 100 and 1000 m isobaths are represented in fine grey lines. The triangle between the Sulawesi and Java Sea indicates the location of the Makassar Strait. Locations Su, Sa, L, D, Ma B, C, D corresponds respectively to the Sulu islands chain, the Savu Sea, the Lifamatola Passage and the Dewakang Sill. Point P refers to the location of the profile which evolution is shown in Fig.3.

Owing to the complex bathymetry of the region, equations are solved on a staggered Arakawa C-grid covering the entire South-East Asian region (see Fig.1) using a rather high regular resolution of about 5 km on the horizontal. This resolution is relatively high compared to previous regional studies (Nugroho et al., 2018; Katavouta et al., 2022) and allows for a better representation of the many narrow straits in the region, especially along the Sunda Islands that separate the Indonesian Seas from the Indian Ocean. To reduce truncature errors of sigma-coordinates while maintaining an accurate description of the complex bathymetry, vanishing quasi-sigma (VQS) coordinates are used on the vertical (Dukhovskoy et al., 2009; Estournel et al., 2021), with 60 levels. Since the coordinates are terrain following, vertical resolution varies across the domain, ranging from around 2 m at the surface to up to 600 m for the last level at the deepest locations.

Currents and tracers are initialized using the $1/12° \times 50$ fields from the Global Ocean Physics Reanalysis from the Copernicus Marine Service (GLORYS12V1, https://doi.org/10.48670/moi-00021), hereafter called GLORYS, interpolated on our higher resolution grid. Daily averages of temperature, salinity, sea surface height, and currents are then provided by the same model and applied at lateral boundaries. Tides are forced by harmonics M2, N2, S2, K2, K1, O1, P1, Q1 and M4 from the tide atlas FES2014 (Lyard et al., 2021). Along with the 5 km resolution, the configuration should be able to resolve between 80 to 90% of the barotropic to baroclinic conversion rate, following Niwa and Hibiya (2011), though the exact location of energy dissipation is still unclear. Surface fluxes are computed from the bulk formulae of Large and Yeager (2004) using variables from the European Centre for Medium Range Weather Forecasts (ECMWF) operational forecasts at $1/8°$ horizontal resolution and 3 hours temporal resolution (https://www.ecmwf.int/). Three hundred and eight rivers are considered in the region. Discharge flux time series are computed every five days and are derived from the 10 km resolution hydrological analysis GLOFAS(Alfieri et al., 2013). More details on the practical implementation of rivers in Symphonie are provided in Nguyen-Duy et al. (2021), Appendix B.

Given the relatively low amount of validation data in the region (Sprintall et al., 2019), simulations have been chosen to coincide with the peak of ARGO floats profiles in the region, and are therefore run on the 2017-2018 period. No ARGO floats were indeed available in two major passages of the ITF before these years, namely the Makassar Strait (linking the Sulawesi Sea to the Java Sea) and the Banda Sea. See Fig.A1 for locations of ARGO floats in the region. Daily averages of each variables are saved for analysis. In total, four simulations have been run - T0, T1, NT0 and NT1 - which details will be described in Sec.3

## 2.3 Advection

In the following, we briefly describe the advection scheme used in Symphonie and the method implemented to improve the diffusive properties of the vertical component. We begin by describing the general framework in which the discussion is set and define the notations used. We then present the advection scheme already in use in Symphonie. Its numerical diffusivity is estimated and compared to physical mixing. A method is finally proposed to improve its diffusive properties and make it usable on the vertical.

### 2.3.1 Notations

As the issue we are dealing with arises from the vertical component of the tracer transport, we will restrict our analysis to a one dimensional situation and use notations considered standard for the vertical. We will also make use of $\Im(\cdot)$ and $\Re(\cdot)$ to denote the imaginary and real part of a complex number, respectively; and *hats* will be used to indicate the Fourier mode representation of a linear operator, e.g. $\hat{f}$ for a given operator $f$.

The continuous advection of a tracer $\tilde{s}$ by an incompressible flow of velocity field $\tilde{w}$ is expressed by the following equation :

$$\partial_t \tilde{s} = -\partial_z(\tilde{w}\tilde{s}) \tag{1}$$

Tilde are used to distinguish continuous fields from their discrete counterparts. On a staggered regular grid $\{z_j\}$ with constant spacing $\Delta z$, discretization of the equation in space in a finite volume formulation for the discrete field $s = \{s_j\}$ advected by a velocity field $w = \{w_j\}$ writes

$$\frac{ds_j}{dt} = -\frac{(F_{j+} - F_{j-})}{\Delta z} \tag{2}$$

where $F_{j+}$ (resp. $F_{j-}$) conceptually represents the value of the flux field $F = \tilde{w}\tilde{s}$ at the interface between cells $j$ and $j+1$ (resp. cells $j-1$ and $j$). In such a formulation, $w_j$ represents the velocity field $\tilde{w}$ sampled at location $z_j - \Delta z/2$ while $s_j$ represents the amount of tracer $\tilde{s}$ in grid cell $j$, ie.

$$s_j(t) = \frac{1}{\Delta z} \int\limits_{z_j - \Delta z/2}^{z_j + \Delta z/2} \tilde{s}(t, z)dx \tag{3}$$

Hence, since $F$ represents the flux at the interface, an estimate $s_{j+}$ of $\tilde{s}(t, z_j + \Delta z/2)$ will have to be reconstructed from the values of $\{s_j\}$ via a given *scheme*. Advection schemes used nowadays in ocean models usually originate from standard interpolations procedures, or weighted (often with non-linear weights) combinations of several other interpolations in order for the computed solution to satisfy certain desired properties.

Note that since quasi-eulerian coordinates are used, the vertical velocity to be considered in practice in the discretization of the vertical advection is actually the *eulerian* velocity $w_j'$, that is the field obtained after removing the vertical displacements of the coordinates due to the moving free surface. Now, for large vertical displacements in the ocean interior caused for instance by strong internal tides, surface displacements are a few order of magnitudes lower than vertical displacements at a few hundred meters depth, and the resulting difference between the true and relative fields is weak there. Thus, and since the theoretical framework used here is rather ideal, we will abusively use $w_j$ (or $W$ when the velocity is considered constant) in the following to refer to this velocity relative to the grid points to simplify the notations.

### 2.3.2 Advection of tracer in Symphonie

Horizontal advection of tracer is carried out using a pseudo-QUICKEST scheme, inspired from the QUICKEST scheme by Leonard (1979), and hereafter referenced to as QKE. Its flux formulation writes formally

$$F^{\text{QKE}} = (1 - (2n_c)^2)F^{\text{UP3}} + (2n_c)^2 F^{\text{UP1}} \tag{4}$$

where $n_c$ is the Courant number on the corresponding direction and $F^{\text{UP1}}$, $F^{\text{UP3}}$ the fluxes for the first order and third order upwind advection schemes, respectively. Following Webb et al. (1998), the UP3 scheme is split into a fourth-order centered advection part ($C_4$) and a bilaplacian diffusive part ($D_4$), and the $D_4$ part is evaluated at time step $t - 1$ during the leapfrog integration, so as to ensure conditional stability. The UP1 part is fully evaluated at time step $t - 1$.

The scheme behaves like a UP3 scheme for low Courant numbers, while for Courant numbers close to 0.5 (i.e. $2n_c \sim 1$), the scheme turns into the forward integration of a first order upwind discretization integrated with time step $2\Delta t$, known to transport perfectly in those conditions. This hybrid behaviour increases the range of stability of the model and allows it to handle more robustly the few hotspots of strong Courant numbers. This scheme has demonstrated its robustness in the past applications of Symphonie, where vertical motions were either relatively low or the size of the domains allowed for a rapid regeneration of water masses by forcing at the boundaries, two assumptions that do not hold anymore in the present configuration.

Vertical advection of tracer in Symphonie is traditionally carried out with a second order centered advection scheme. However, under strong vertical motions, dispersive errors can have spurious effects on the density field, especially on the long run (see e.g. Griffies et al., 2000; Hecht, 2010) and some kind of numerical diffusion is therefore required to prevent small scales density instabilities from developing, either by introducing linear diffusion operators (often implicitly included in upwind formulations) or by using non-linear limiting procedures such as TVD schemes (Cushman-Roisin and Beckers, 2011) or their stronger version, FCT schemes (Zalesak, 1979), in order to enforce some kind of monotonicity in the evolution of the field. We opted for the first approach by building on a linear scheme already implemented in Symphonie and trying to improve its diffusive properties through a lightweight formulation to make it usable on the vertical at rather low additional computational cost.

### 2.3.3 Excessive diffusion on the vertical

We can simplify the analytical discussion by considering that along the vertical, Eq.4 essentially boils down to the formulation of a UP3 scheme in the regions of interest of the model. Indeed, using typical numerical values for internal tides of $W \approx 10^{-3} - 10^{-4}$ m.s$^{-1}$, $\Delta z \approx 10^1 - 10^2$ m and a time-step $\Delta t = 180$ s $\approx 2 \times 10^2$ s, we get $n_c \approx 10^{-2} - 10^{-5}$, so that $n_c \ll (1 - (2n_c)^2) \sim 1$ and the role played by the UP1 part can be neglected, at least in the ocean interior. Though of higher order than the UP1, the UP3 scheme is nonetheless known to be still excessively diffusive (see e.g. Madec et al., 2022).

To quantitatively illustrate this behaviour, we derive the dispersion relation for a single Fourier mode $s_j \propto s_k e^{i(kj\Delta z - \omega t)}$, linking $\omega$ to $k$ and expressing how a given harmonic wave with wavenumber $k$ evolves over time. In a such a spectral framework,

the diffusive aspect of a scheme is expressed by the addition of a damping term in the corresponding dispersion relation. More specifically, writing this dispersion relation in its canonical form $\omega(k) = \Omega(k) - i\gamma(k)$, with $\Omega$ and $\gamma$ real valued functions of $k$, the scheme is said to be diffusive when the damping coefficient $\gamma(k)$, that quantifies the typical time $\tau = 1/\gamma$ at which a given wavelength is being damped out, is strictly positive.

If the scheme can be split as the sum of a purely dispersive part $\mathcal{A}[s]_j$ and a purely dissipative part $\mathcal{D}[s]_j$, i.e.

$$\frac{d}{dt}[s_j] = \mathcal{A}[s]_j + \mathcal{D}[s]_j \tag{5}$$

it follows immediately that $\Omega(k) = -\Im(\hat{\mathcal{A}}(k))$ and $\gamma(k) = -\Re(\hat{\mathcal{D}}(k))$, as the pure dispersion and pure dissipation assumptions ensure that both $\Re(\hat{\mathcal{A}}[k])$ and $\Im(\hat{\mathcal{D}}[k])$ are zero (see e.g. Webb et al. (1998) or Soufflet et al. (2016) for more details).

Let us now focus more specifically on the damping coefficient obtained for the UP3 scheme. The full discretization of this scheme can be found in both Webb et al. (1998) and Marchesiello et al. (2009). Here, only its dissipative parts $D_4$ is needed, and it writes

$$D_4[s]_j = \frac{1}{12}\frac{|W|}{\Delta z}(-s_{j+2} + 4s_{j+1} - 6s_j + 4s_{j-1} - s_{j-2}) \tag{6}$$

Replacing each $s_j$ by its Fourier mode value, assuming for simplicity the vertical speed $W$ to be constant and using the normalized form $\theta = k\Delta z$ for the wavenumber, we have

$$\hat{D}_4(\theta) = -\frac{W}{6\Delta z}(3 + \cos 2\theta - 4\cos\theta) \tag{7}$$

This finally leads to the most compact form of the damping coefficient

$$\gamma^{\text{UP3}}(\theta) = \frac{W}{3\Delta z}(1 - \cos\theta)^2 \tag{8}$$

For the largest wavelengths, i.e. $\theta \to 0$, we have $\gamma^{\text{UP3}}(\theta) \to 0$, which is expected since we do not want the largest scales to be damped out; while for the smallest wavelengths, i.e. $\theta \to \pi$, the damping is maximal and equal to $\frac{4W}{3\Delta z}$ : the smallest, noisy, wavelengths are damped out. Comparing this to the damping coefficient obtained for the simplest physical diffusion term $\kappa\partial_{zz}s$, $\kappa$ being the vertical diffusion coefficient computed from the turbulence closure scheme, discretized in its usual form $\kappa(s_{j+1} - 2s_j + s_{j-1})/(\Delta z)^2$ that leads to a physical damping coefficient

$$\gamma^{\text{phy}}(\theta) = \frac{2\kappa}{\Delta z^2}(1 - \cos\theta) \tag{9}$$

the ratio of numerical to physical damping $\Gamma_{\text{UP3}} = \gamma^{\text{UP3}}/\gamma^{\text{phy}}$ writes

$$\Gamma_{\text{UP3}} = \frac{W\Delta z}{6\kappa}(1 - \cos\theta) = \frac{P_e}{6}(1 - \cos\theta) \tag{10}$$

where $P_e$ is the grid Peclet number. Typical values of $P_e$ found in the model can be obtained using the same numerical values as before and considering a value of $\kappa \approx 10^{-5}$ m$^2$.s$^{-1}$ typical for vertical diffusivity in the ocean interior (e.g. Polzin et al., 1997; Alford et al., 1999). This leads to $P_e \approx 10^2 - 10^4$. For intermediate wavelengths $N = 2\pi/\theta$ of a ten of grid points,

that is $\theta = 2\pi/10$, typical of the number of grid points used to represent a thermocline, we have then $\Gamma_{\text{UP3}} \approx 3 - 300$. This means that the numerical mixing resulting from the discrete vertical advection is at least three times larger than the physical mixing at this scale, even under relatively conservative assumptions about the quantitative values, and can even be two orders of magnitude larger. The numerical diffusion is thus not *selective enough*, in the sense that it spuriously damps physical scales.

In the following, we propose a method to improve this selectivity.

### 2.3.4   Filtered diffusion

As discussed previously, UP3 can be written as the sum of a purely dispersive ($C_4$) and a purely dissipative ($D_4$) components. Since the damping originates from $D_4$, we now aim at making this term more scale-selective. This can be achieved through a filtering process : assuming a high-pass filter $\Phi$ that applies on the tracer field $s$, the idea is to apply the diffusive operator on

the filtered field only, i.e. turn $D_4[s]$ into $D_4[\Phi[s]]$, so that only the high spatial frequencies (smaller scales) are dissipated. In spectral space, since $D_4$ is linear, the damping coefficient for this scheme becomes formally

$$\gamma(\theta) = -\Re(\hat{D}_4(\theta))\hat{\Phi}(\theta) \tag{11}$$

Therefore, for large scales where $\hat{\Phi}(\theta) \sim 0$, $\gamma(\theta)$ will be almost zero, while for small scales where $\hat{\Phi}(\theta) \sim 1$, $\gamma(\theta)$ will remain unchanged. We assumed here $\hat{\Phi}(\theta)$ to be real valued, an assumption justified by the fact that it would otherwise

introduce spurious lag in the filtered signal. Theoretically, many filters could fit. However, we now focus on the specific filter with which the method has been primarily developed and other formulations will be discussed in Sec.4. Inspired by a formalism proposed in Juricke et al. (2020b) in the context of the momentum equation, we define a filter $\Phi = \mathbf{1} - \phi$ where $\phi$ is a *low*-pass filter and $\mathbf{1}$ the identity operator. Formally, we rewrite the formulation of the UP3 in what will be referred to later as the *filtered formulation* :

$$\text{UP3}[s] = C_4[s] + D_4[s] - D_4 \cdot \phi[s] \tag{12}$$

Eq.11 then becomes

$$\gamma(\theta) = -\Re(\hat{D}_4(\theta))(1 - \hat{\phi}(\theta)) \tag{13}$$

Note that the original formulation of Juricke et al. (2020b) makes use of a more general filtering, namely

$$\Phi_\alpha = \mathbf{1} - \alpha\phi \tag{14}$$

where $\alpha > 0$ is a parameter. $\alpha$ allows to control the strength of the filtering, $\alpha = 0$ corresponding to a no-filter situation while $\alpha = 1$ corresponds to a fully active filter. The latter corresponds to our filtered formulation. The interesting aspect of the original formulation lies in the use of a value $\alpha > 1$. Such a regime could be interesting for the tracer equation, but since it comes with several issues, we restrict ourselves in this paper to the simpler case $\alpha = 1$.

For our practical analysis, we introduce the three-points filter $\phi_3[s]_j = (s_{j+1} + 2s_j + s_{j-1})/4$. If any low-pass filter can be

chosen in theory, we have used this one for several reasons : it is the same as the one used in Juricke et al. (2020a); its simplicity

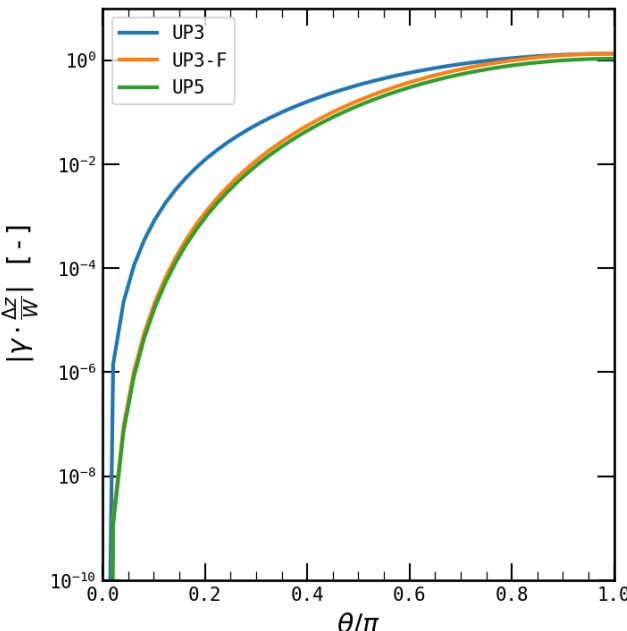

**Figure 2.** Comparison of normalized damping coefficients $\gamma W/\Delta z$ as a function of $\theta/\pi$ for several advection schemes. The y-scale is logarithmic.

allows for straightforward theoretical developments; and finally its limited stencil size does not increase the computational cost of the scheme too much (the full stencil size for computation of the advection term increases from 7 to 9 grid points), which may be desirable if the model is to be run over long time periods. This choice is further discussed in Sec.4. Its spectral representation writes $\hat{\phi}_3(\theta) = (1+\cos\theta)/2$. This leads to the following damping coefficient for the filtered UP3 scheme (hereafter UP3-F) :

$$\gamma^{\text{UP3-F}}(\theta) = \frac{W}{3\Delta z}(1-\cos\theta)^2(1-\frac{1}{2}(1+\cos\theta)) = \frac{W}{6\Delta z}(1-\cos\theta)^3 \tag{15}$$

The relative damping for the filtered formulation $\Gamma_{\text{UP3-F}} = \gamma^{\text{UP3-F}}/\gamma^{\text{phy}}$ becomes therefore

$$\Gamma_{\text{UP3-F}} = \frac{P_e}{12}(1-\cos\theta)^2 \tag{16}$$

Comparison of $\gamma^{\text{UP3}}$ and $\gamma^{\text{UP3-F}}$ are provided in Fig.2 Considering again a wavelength of 10 grid points and using the same numerical values as before, we have $\Gamma_{\text{UP3-F}} \approx 0.3-30$. The damping at large scales as been reduced by one order of magnitude, and for the range of $P_e$ of concern, the numerical dissipation can be now as high as or lower than the physical one. Numerical diffusion will now exceed physical diffusion were the vertical grid spacing is the coarsest, that is at depth. Indeed, $\Delta z = 100$ m is reached at 1000 m depth in our model, and at this depth, tracer profiles are already smooth enough so that their second or higher order derivatives are almost zero and diffusion, even numerical, is not really active at all. We therefore consider those values as being *a priori* acceptable. The scheme finally used for vertical tracer equation will be referred to as the QKE-F

scheme. The flux evaluation in Eq.4 is thus finally modified as such :

$$F^{\text{QKE}} = (1 - (2n_c)^2)F^{\text{UP3-F}} + (2n_c)^2 F^{\text{UP1}} \tag{17}$$

The CFL stability condition of the resulting scheme is not modified. Indeed, following Lemarié et al. (2015), the CFL constraint of the UP3 integrated as in our simulations using a leapfrog time-stepping with Robert-Asselin filtering and coefficient taken equal to 0.1 and lagging the evaluation of the diffusive term in time is $n_c \leq 0.472$. Using the same method, we can show that 290 the constraint for UP3-F is $n_c \leq 0.507$. Since the UP1 term is lagged in time, it is as if it was integrated using a forward time stepping with time step $2\Delta t$, and the CFL constraint for this term is 0.5, close to 0.472 and 0.507. Now, for $n_c \sim 0.5$, the UP1 term largely dominates in QKE and sets therefore the stability constraint, which is thus $n_c \leq 0.5$ for both QKE and QKE-F, as confirmed by a more formal stability analysis (not shown).

## 3 Results

A simplified one-dimensional framework such as the one presented above is essential for gaining insight into the actual behaviour of a numerical method (Griffies et al., 2000). However, this theoretical development relies on limiting assumptions intended to maintain analytical simplicity and, therefore, cannot be the only basis for designing practical advection schemes. This section provides a real case study for the validation of the method, illustrated as a way to improve water masses representation in simulations of the SEA Seas using the Symphonie model. We start by discussing briefly the way spurious mixing will be 300 diagnosed in the model. A presentation of the results obtained from simulations with and without filtering is then presented and compared to their counterparts without tidal mixing. Doing so, we illustrate clearly the negative impact of numerical mixing and the improvement offered by our method in terms of representations of the water masses.

### 3.1 Diagnosing numerical mixing

Several frameworks for diagnosing numerical mixing have been proposed in the literature (e.g. Griffies et al., 2000; Lee et al., 305 2002; Burchard and Rennau, 2008; Gibson et al., 2017), each with their own strengths and weaknesses. These frameworks are however designed for inter-comparisons in a theoretical context of numerical methods for which the analytical computation of diffusion is not always possible, such as for non-linear schemes. As we focus here on the practical application of the method, we choose here to proceed in a similar way as in Marchesiello et al. (2009) and to use anomalies in tracer fields (especially salinity, which exhibits a complex vertical structure with local maxima and minima) as a proxy for spurious mixing.
Several datasets are used for validation, namely GLORYS reanalysis, ARGO profiles (Wong et al., 2020) as well as OSTIA SST (Donlon et al., 2012) and SMAP/SMOS sea surface salinity (Kolodziejczyk et al., 2021) gridded products. Details on the data and processing are given when first used.

## 3.2 Effect of the QUICKEST scheme without the filtered formulation

To explicitly illustrate the excessive diffusivity of the QKE scheme on the vertical when used along with tidal forcing as well
as the improvements brought by the filtered formulation, two reference simulations are first run with the filtered formulation
turned off ($\alpha = 0$, equivalent to using QKE in its simplest formulation): one with (T0) and one without tides (NT0).

### 3.2.1 Salinity and temperatures profiles

Comparison to GLORYS data is first carried out to assess the ability of the model to maintain the salinity field in the South
China Sea (SCS) over time (Fig.3). Considering the higher availability of ARGO floats in this basin (see Appendix A), it is
assumed that GLORYS data offer a good and easy-to-handle reference for the state of the ocean. To allow for a more compact
representation, the focus is laid on the $50 - 350$ m layer, were differences between simulations are the most salient. Simply
using the QKE scheme leads to an expected unrealistical erosion of the salinity maxima observed in Pacific waters in the
northern South China Sea over the course of the first months of simulation in the T0 simulation (forced with tides). The salinity
maximum of about $34.7$ psu in the $100 - 200$ m layer inherited from the initialization with GLORYS and visible at the very
beginning of the simulation disappears to form a nearly isohaline profile below $100$ m after a few months, with salinity below
$34.6$ psu. On the other hand, NT0 simulation is able to maintain the salinity peak, comparable to GLORYS reference.

Secondly, comparison of mean temperature and salinity profiles to ARGO data available in the region over the period of
interest are carried out (Fig.4). For each simulation, a dataset of simulation profiles collocated to the ARGO profiles in the
zone - in terms both of space and time - is built. The profiles are split in several clusters corresponding roughly to the major
basins sampled in the zone (see Fig.A1). They are then interpolated on the same grid over the first $500$ m and a mean profile is
finally computed for each simulation and each cluster for comparison. The focus is laid on the first $500$ m where the temperature
and salinity profiles exhibit their richest spatial structure, and as the ITF is also mostly located in this layer. In Fig.4, we display
only profiles in the Molucca Sea for clarity (see Fig.1), since the results in this basin are representative of the overall trend.
However, the same analysis has been conducted in each basin of the Southeast Asia region and will be discussed later on in the
light of other simulations.

As suggested by the drift observed for the South China Sea profile, the mean salinity between $100$ m and $300$ m in T0 (light
grey line, panel (b)) is strongly underestimated with respect to ARGO data, with a bias around $-0.15$ psu at $125$ m depth. The
salinity maxima found between $100 - 200$ m in observations and characteristic from Pacific Waters has completely disappeared.
In panel (a), the temperature gradient in the 10-300 m is also slightly less steep, and the temperature in the upper $100$ m in the
T0 simulation also displays a negative bias of up to $2°$ C at the surface.

Conversely, the salinity bias in the thermocline for the simulation without tides NT0 (light orange) with respect to observa-
tions is positive, and the salinity maximum at $120$ m is overestimated by about $0.1 - 0.2$ psu. Differences in surface salinity are
more complex to analyse and are discussed in more details in Sec.3.4. Differences in the temperature profiles are less obvious
than for T0, even though a slight positive bias can be observed at the surface. Those differences result from an underestimation
of physical mixing in NT0, that results in a misrepresentation of water masses in the region : as tides and the associated internal

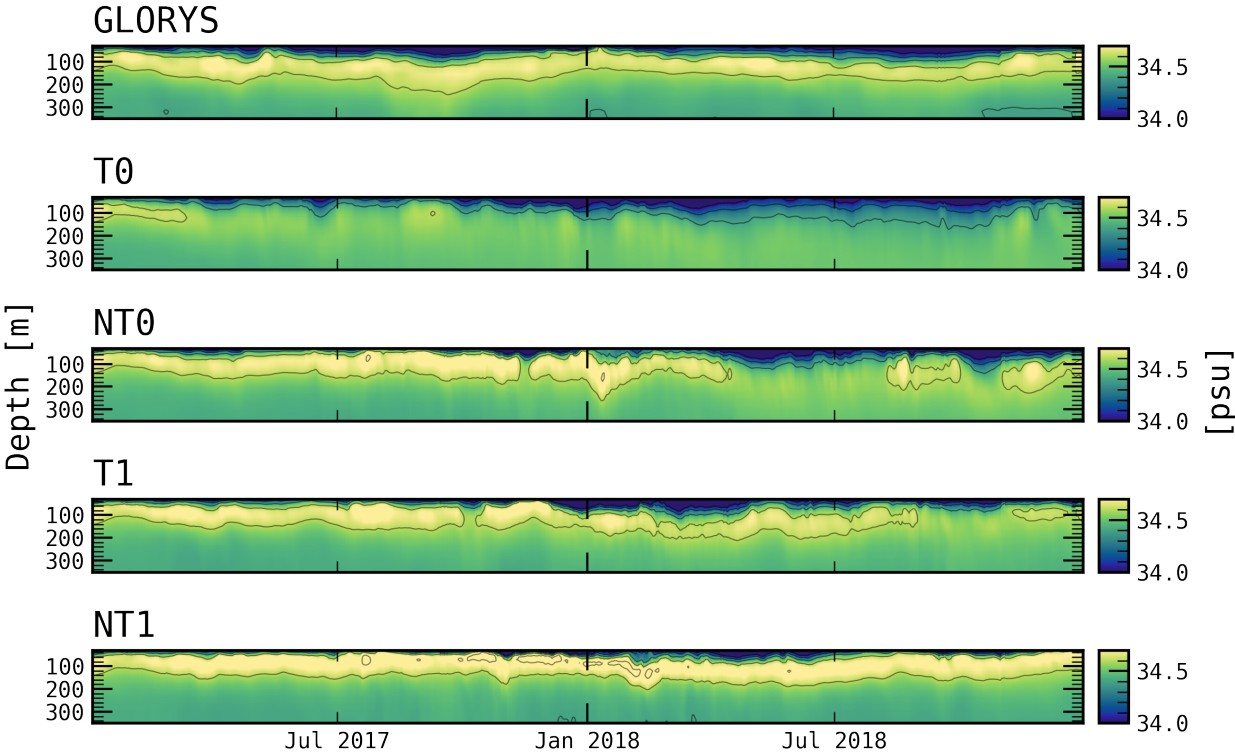

**Figure 3.** Temporal evolution of a salinity profile $(50 - 350$ m) averaged over a 1° box in the South China Sea, at location P in Fig.1, in both GLORYS reanalysis and several Symphonie simulations. Contours are plotted every 0.2 psu

waves field are not taken into account, highly saline waters coming from the Pacific are not well mixed, which biases the model output.

### 3.2.2 Comparison to satellite sea surface temperature data

Finally, the *Operational Sea Surface Temperature and Sea Ice Analysis* (OSTIA) Level 4 Sea Surface Temperature daily
product is also used for comparison. This product has been obtained by mapping existing observations from several instruments on a regular 1/20° grid with optimal interpolation, and daily averaging results. Its resolution is therefore comparable in both space and time to our simulations output. As with every highly processed product, care has to be taken not to put excessive confidence in such data, since biases can exist, especially on small coastal features (see e.g. To Duy et al., 2022). Large scale trends are however well captured by OSTIA, and comparison is therefore carried out by computing the mean bias over the
entire simulation period between simulations and OSTIA. Results are displayed in Fig.5. Since errors in the Pacific and Indian Ocean are more likely caused by open-boundaries forcing, the focus is laid on the interior basins, where most of the differences are concentrated.

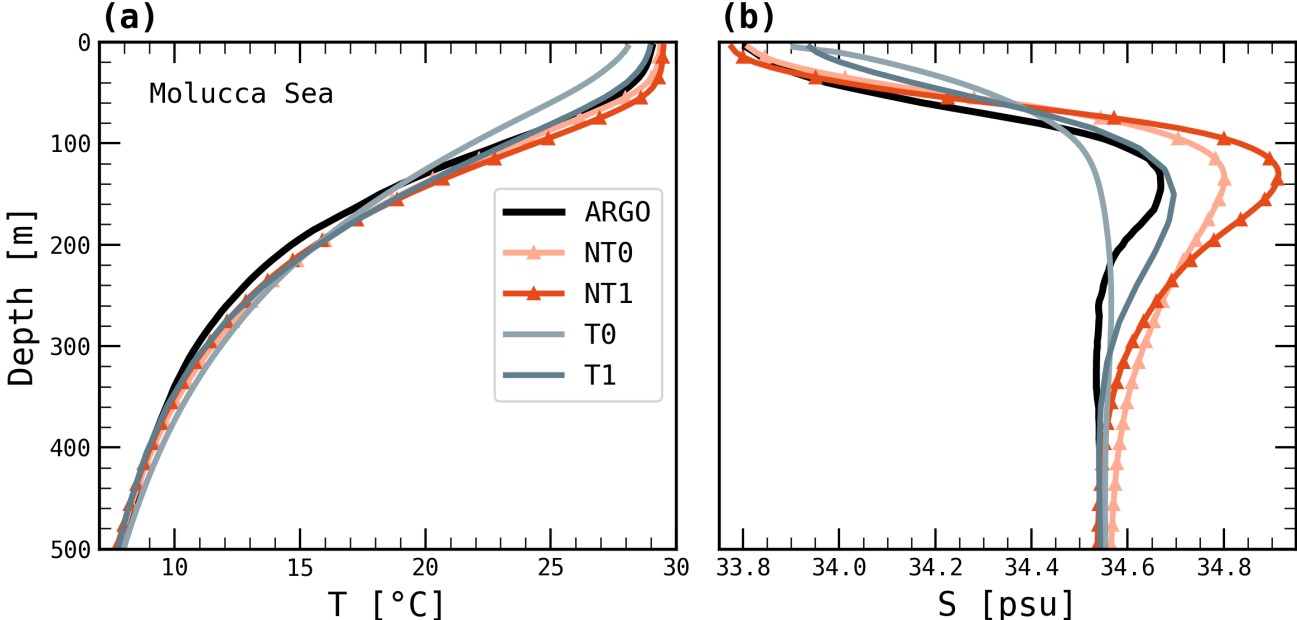

**Figure 4.** Mean (a) temperature and (b) salinity profiles over the first 500 m in the Molucca Sea for simulations T0, NT0, T1, NT1 and ARGO profiles.

A cold negative bias of up to $2°$ C spans the entire Indonesian Seas in simulation T0 (bottom left panel), especially marked over the Savu Sea, the Dewakang Sill as well as along the islands south of the Sulu Sea (see Fig.1, top left panel for details on the locations). Conversely, NT0 displays a warm positive large scale SST bias of up to $1°$ C with respect to the reference dataset, with maximum positive biases at the same locations where maximum negative biases are observed in T0. Both locations are known for being strong internal waves generation sites (Apel et al., 1985; Nagai and Hibiya, 2020) and exhibiting strong SST variations due to internal tides activity (Ray and Susanto, 2016). This confirms the conclusions obtained by examining salinity profiles : the (numerical) mixing is too strong in the presence of tides in T0, bringing colder water from below to the surface, thus cooling the simulated SST; conversely, the (physical) mixing is overly too weak in NT0 due to the absence of tides-induced mixing, therefore preventing warmer surface waters to be mixed with colder waters at depth.

### 3.3 Effect of the QKE-F scheme

To assess the effect of the filtered formulation, two new simulations are run with the filtered formulation turned on ($\alpha = 1$) : one including tides (T1), along with its counterpart without tides (NT1). Differences with T0 and NT0 simulations on the same diagnostics as described above are presented in the following section. The main differences between the various simulations are summarized in Tab.1.

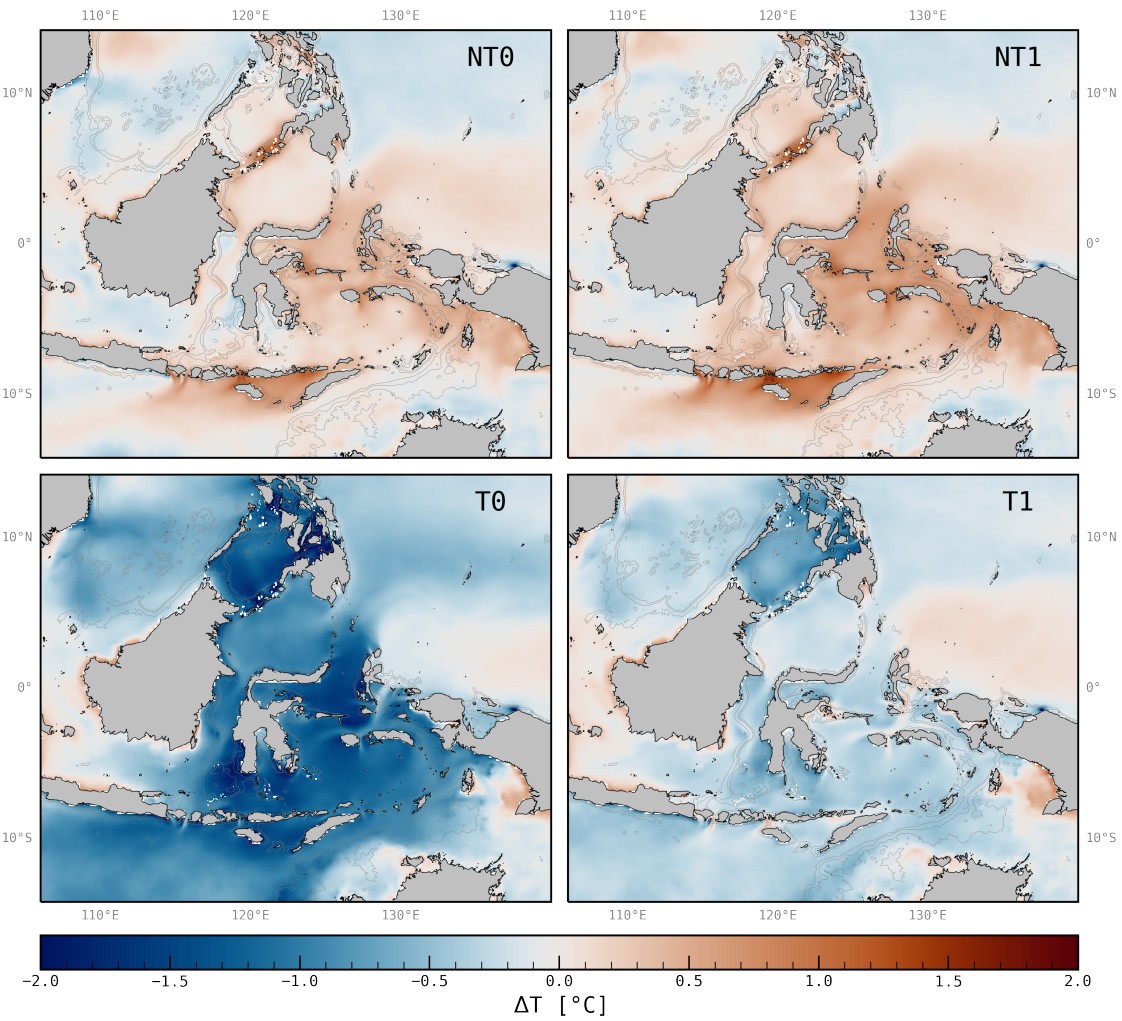

**Figure 5.** Mean SST bias with respect to the OSTIA dataset ($\Delta T = SST(SYMPHONIE) - SST(OSTIA)$) for simulations T0, NT0, T1, NT1, computed over years 2017-2018. Positive values correspond to an overestimation of model output. The 100, 500 and 1000 m isobaths are also displayed (fine grey lines).

|  | QKE | QKE-F |
|---|---|---|
| Tides | T0 | T1 |
| No tides | NT0 | NT1 |

**Table 1.** Summary of all the simulations run in this study along with their main differences. Columns correspond to the advection scheme used, i.e. either QKE or QKE-F, while rows indicate if tidal forcing was used or not.

### 3.3.1  Salinity and temperature profiles

Looking at the evolution of the salinity profile in the South China Sea in Fig.3, the salinity maximum in T1 has been restored with respect to T0, and the evolution of the profile over the course of the T1 simulation lies much closer to GLORYS reference dataset, but also to NT0. This implies first that tides do not play a significant role in the (trans)formation of water masses at this location; and second that the difference between T0 and NT0 is caused by the excessive numerical diffusion occurring when tides are added. Concerning simulations without tides, the salinity evolution for NT1 depicted in Fig.3 is also qualitatively better than for NT0, implying that there is still spurious mixing happening in NT0 in the South China Sea. This is to be expected, since even simulations without tides can display transient relatively high vertical velocities (on the order of a few tens of meter per day) and therefore spurious mixing, especially at eddy-resolving resolution (see Megann et al., 2022).

Results in terms of mean profiles in the Molucca Sea are also reported in Fig.4 for T1 and NT1. T1 performs better than T0 and overall follows the ARGO reference more closely: the negative $\sim 1°C$ temperature bias at the surface in T0 and the negative $\sim 0.1$ psu salinity bias in thermocline waters disappears in T1. The implemented filtered formulation has thus effectively reduced the spurious mixing developing in the context of strong tidal forcing down to acceptable levels. The picture is different for simulations without tides, NT0 and NT1. In terms of salinity, NT1 performs worse than the other simulations, even when compared to NT0, with an increased positive bias of 0.3 psu at 120 m depth, against $\sim -0.15$, 0.01 and $+0.2$ psu for T0, T1 and NT0, respectively. Tidal mixing indeed takes place in the Molucca Sea and drives the transformation of water masses (see e.g. Koch-Larrouy et al., 2007). The weaker bias from NT0 in comparison to NT1 is due to the fact that the spurious numerical mixing in NT0 is higher than in NT1, thus making up for a part of the lacking tides-driven physical mixing. This shows again that even without tidal motions, a sensible amount of spurious mixing is still taking place in NT0. With fewer numerical mixing in NT1, the underestimation of physical mixing is made all the more salient, which increases the bias and overestimation of the salinity maximum in NT1 compared to all other simulations.

To provide a more synoptic picture of the effectiveness of the filtered scheme, we carry out a comparison of all simulations in every basins of the zone. For a given simulation and zone we define an integrated metric of the performances in terms of water mass representation, to allow for a more direct and quantitative comparison than through visual inspection solely. In each zone, a weighted root mean squared error between each simulation and the reference ARGO profile is computed for the mean profile.

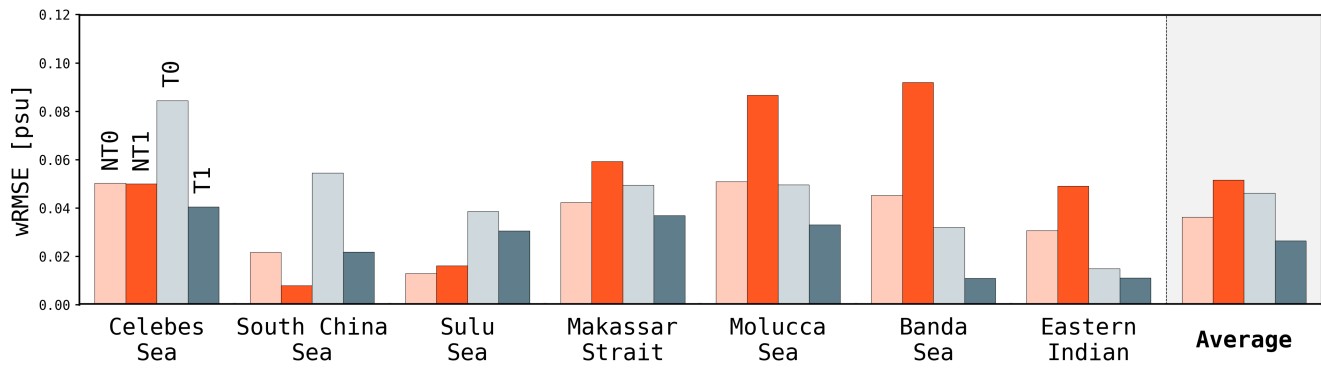

**Figure 6.** wRMSE between mean salinity profiles for simulations T0, NT0, T1 and NT1 (for each zone, left to right bar respectively) for each of the ARGO subgroups. The last column displays the metric averaged over all zones.

As the variations in the upper layers are of higher magnitude than in the deeper layers, we weight accordingly the error by reducing the importance given to values where the variation is higher. The resulting metric is referred to as wRMSE, and more details on its computation are provided in Appendix B. Here, the focus is laid on salinity as the effect of numerical mixing on these profiles is more obvious (see Fig.4), but the same computations have been made for the temperature with similar results (not shown).

Results of wRMSE computations on the salinity profiles for all simulations output for the various zones are displayed in Fig.6. Values lie between $0.01$ and $0.1$ psu. Comparing the integrated wRMSE scores in the Molucca Sea with the profiles provided in Fig.4, our metric indeed quantitatively translates the qualitative observation that in this zone, T1 performs the best and NT1 performs the worst while T0 and NT0 are in-between.

Comparison between T0 and T1 shows systematically improved wRMSE in T1 in each zone, with much greater difference in zones directly neighbouring the Pacific (Celebes and SCS) : around $0.04$ psu in the SCS, and only $0.01$ psu in the Makassar Strait. Except for the Sulu Sea, T1 also improves the results with respect to NT0. The difference is however usually smaller, except in the exit basins (Bandas Sea and Indian Ocean) where the difference is sensible : wRMSE for T1 is $0.02 - 0.03$ psu lower than for NT0 in those basins.

Performances of NT1 relatively to the other simulations greatly vary with the location of the zone along the course of the ITF, that is from the Pacific to the Indian Ocean through the Celebes Sea, the Makassar Strait and the Banda Sea. NT1 performances in the basins of the downwind of the ITF (Makassar strait, Banda Sea, Eastern Indian), where tidal mixing plays an important role, are by far the worse : around $0.08$ psu between T1 and NT1 in the Banda Sea, and $0.4$ psu in the Eastern Indian Ocean. However, in the SCS, NT1 performs better than all the others. The reason behind this best performance of NT1 in the SCS is still unclear, and might be linked to some other factors, including the details of the configuration. Similarly, best performances of NT0 and NT1 in the Sulu Sea are somewhat unexpected, and this feature will be discussed below in the light of comparison to satellite data.

Figure 6 also displays the average over all basins of the wRMSE for each simulation, with a uniform weighting for each zone. Using this SEA average metric, T1 performs overall best than the others, while NT1 performs the worst. The wRMSE could have also been computed for each simulation by comparing the average profile over all available profiles over the ensemble of the basins to the same profile obtained with ARGO data; however, this would have led the results to be strongly biased towards zones with the most data - in particular the SCS. The underlying assumptions here are that the amount of profiles available in each basin is sufficiently high so that comparisons can be trusted equally and that the correct representation of each basin is of equal importance.

### 3.3.2 Comparison to sea surface temperature data

Finally, we also validate the filtered formulation against OSTIA SST data. The mean SST in T1 is greatly improved compared to T0 (Fig. 5) : the negative bias over the SEA Seas is lower by one order of magnitude, and is now on the same order as errors made in the Pacific Oceans, that is a few tenths of Celsius. This further confirms that our filtered formulation improves the representation of sea surface temperature in the context of strong tidal mixing.

The SST in the Sulu and Philippine Seas however still exhibits a strong cold bias in T1, up to $1°$ C locally, weaker than in T0 but stronger than in NT0 and NT1. The explanation for this behaviour has been traced to an overestimation of tidal currents in this basin due to an imperfect representation of the numerous islands and channels around it, especially in the Philippine Archipelago. Manual modification of the land mask to improve the representation of the zone had been carried out, with sensible improvements when compared to the first tests, but tidal amplitudes remain too strong there. This results in an overestimation of the naturally strong internal tide activity in the area (Apel et al., 1985), which can only dissipate within the basin due to its enclosed nature, thus increasing the mixing. As a result, tidal vertical mixing is overestimated there, leading to spurious damping of the profiles and SST underestimation. This is slightly "corrected" when the filtering is activated (T1 vs. T0), and even greatly "corrected" when tides are turned off (NT0 vs. T0, NT1 vs. T1).

Besides, the previously described positive SST biases in NT0 simulation shown in Fig.5 are slightly increased in NT1, indicative of weaker vertical mixing in the interior basins - namely along the Sulu islands chain, the Savu Sea as well as around the Lifamatola Passage. Those results are again related to a reduction in spurious numerical mixing induced by the filtered scheme in NT1 compared to NT0, that partly compensates for the lack of tides-induced *physical* mixing in NT0, and makes it more salient in NT1.

### 3.4 Non-linear effect of spurious numerical mixing on sea surface salinity

The results concerning numerical mixing presented so far - higher SST and strong erosion of the salinity maxima - can be explained in the simple framework of one dimensional vertical diffusion, interpreting spurious numerical mixing as simply a higher diffusion, smoothing gradients and mixing properties throughout the column. In a realistic ocean model however, many components interact, often in non-linear ways, and simple errors generated by one component can result in *a priori* unexpected behaviours. We discuss briefly an example of this in the case of air-sea interactions by comparing simulations output to sea surface salinity (SSS) products obtained from the SMOS and SMAP missions. The dataset used is the SSS SMOS/SMAP OI

L4 product, such as described in Kolodziejczyk et al. (2021). Symphonie simulations output are downsampled at the same weekly frequency and interpolated on the same 25 km resolution grid. Results are displayed in Fig.7.

A thorough discussion of all the biases in the region is out of scope of this study, and we will comment instead on the large scales pattern. Overall, the bias is mostly lower than $\pm 0.25$ psu, but can be locally increased up to $\pm 1$ psu. A common pattern throughout all simulations is that several coastal regions exhibits large fresh biases, especially around Borneo, resulting from overestimated river discharges.

Differences between NT0 and NT1 are negligible, and fall within the error margin of such highly processed SSS products, especially in a region such as SEA, with its numerous islands and complex coastal features. Unlike for SST, which was overestimated in both those simulations due to a lack of physical mixing, SSS bias doesn't exhibit any easily distinguishable pattern in simulation without tides.

As for simulations with tides, a large fresh bias spans the entire Indonesian seas region in T0, and is sensibly increased at locations of increased tidal mixing (locations Su, L, Sa in Fig.1), reaching negative salinity biases of up to 1 psu. This bias varies seasonally, with a large intensification of the pattern north of the equator during boreal summer and south of the equator for boreal winter (not shown).

Those differences are in fact the result of the air-sea interactions. Since surface fluxes, including latent heat flux (LH), are computed from bulk formulae, they adjust to the surface fields of the model - classically only SST - in a highly non-linear fashion caused notably by atmosphere stability considerations (see Large and Yeager, 2004). More precisely, the freshwater input at the surface in the absence of rivers is the difference between evaporation and precipitations. Precipitations are prescribed directly and do not change throughout the various simulations. Conversely, evaporation is proportional to the LH which is itself a complex function of the forced atmospheric variables and the SST (which, as previously shown in Fig.5, varies between the simulations). Mean values over the interior basins of LH as computed by the model for the various simulations are displayed in Fig.8. Since LH is by convention positive towards the ocean and usually negative, lower absolute values of LH implies less loss of energy from the ocean to the atmosphere, and therefore less evaporation. As expected, LH values for NT0 and NT1 are quite similar, though the mean in NT1 is slightly higher, due to the slightly higher SST (see Sec.3.3.2). In T0, the absolute LH is considerably lower than in other simulations ($\sim 40$ W.m$^2$ lower than NT1). Lower latent heat flux leading to a much weaker evaporation in T0 thus induces an excess of freshwater atmospheric input in the region, and explains the observed bias. Nevertheless, this mechanism alone cannot explain all the differences. Indeed, though the LH is also slightly weaker in T1 with respect to NT0 and NT1, surface waters in T1 are slightly too salty, though the magnitude of the bias is much lower than the one in T0. This results from the intensified numerical mixing caused by internal tides, even when QKE-F is used, bringing saltier water from deeper levels to the surface, in a similar fashion as for the SST. This tidal mixing induced salinity input compensates the lower evaporation caused by colder surface waters in T0 (see Fig.5).

A precise diagnostic aiming at distinguishing between atmospheric fluxes and diffusive fluxes would need to be carried out in order to draw more precise conclusions on their relative roles. Such a diagnostic is however not possible with daily averaged output and would require the implementation of online diagnostics as well as running new simulations, a task that has not been carried out in the scope of this study.

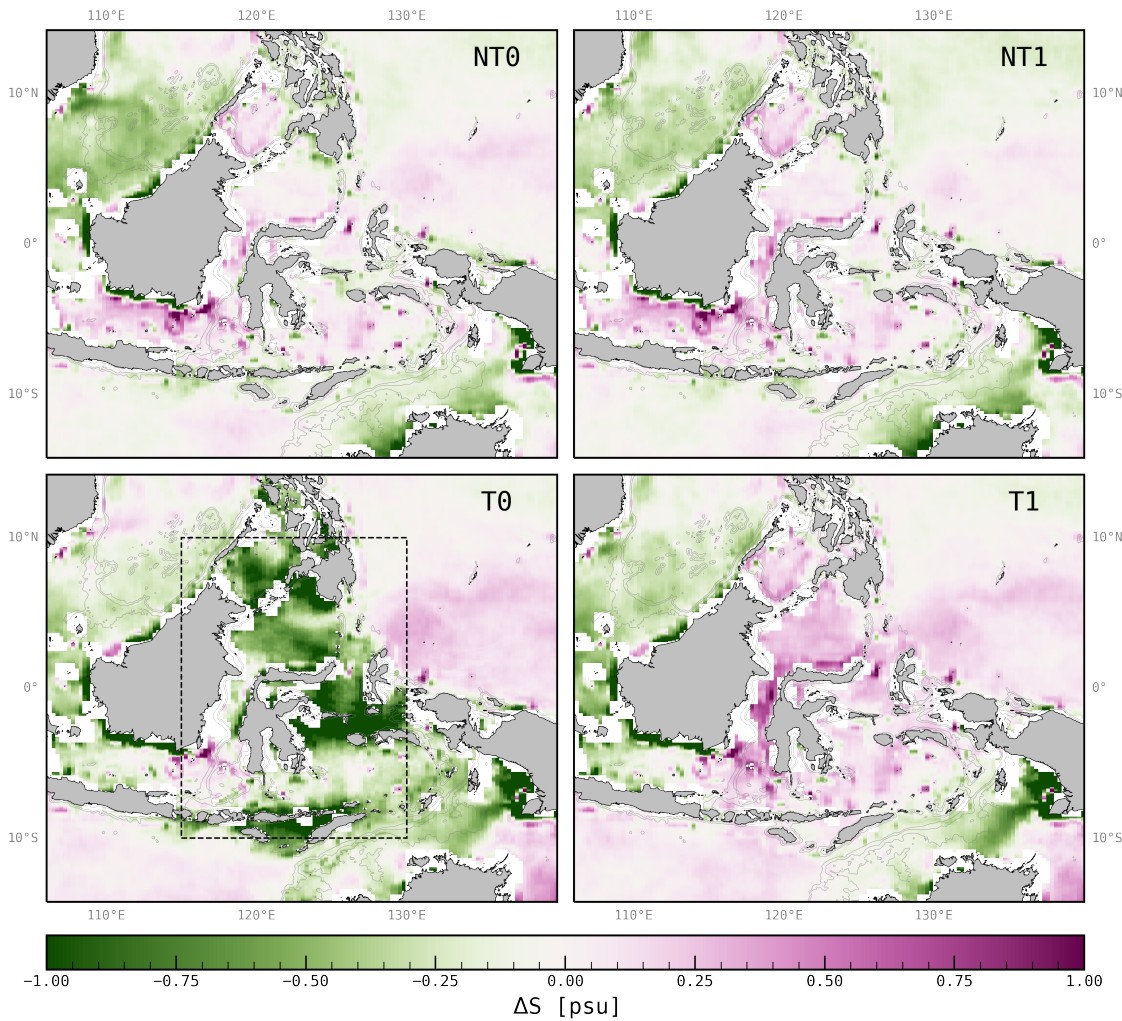

**Figure 7.** Mean SSS bias with respect to the SMAP/SMOS dataset ($\Delta S = $ SSS(SYMPHONIE) $-$ SST(SMAP/SMOS)) for simulations T0, NT0, T1, NT1, computed over years 2017-2018. Positive values correspond to an overestimation of model output. The 100, 500 and 1000 m isobaths are also displayed (fine grey lines).

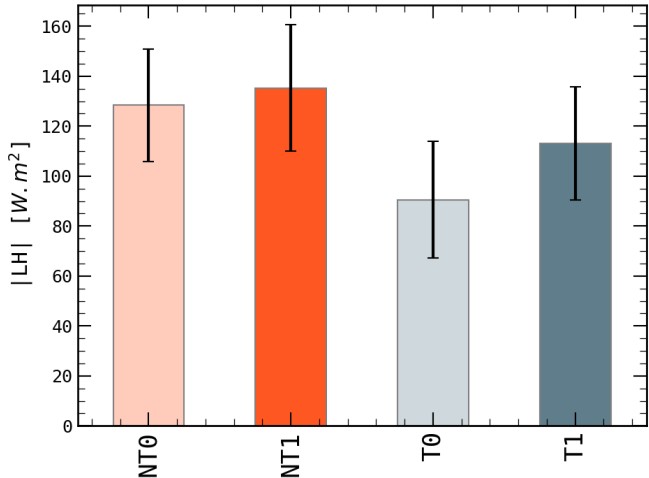

**Figure 8.** Absolute mean latent heat flux (in W.m$^{-2}$) computed over the zone defined by the square in Fig.7 for all the simulations. Error-bars correspond to 1 standard deviation, computed over monthly averaged values and the same spatial domain.

## 4 Discussion

Comparisons of a set of four sensitivity simulations, with and without tides and with and without activation of the filtered formulation, with available ARGO temperature and salinity profiles and OSTIA sea surface temperature data showed that the filtered formulation brings numerical diffusion down to a level allowing for a correct representation of water masses in our simulations over the Southeast Asia Seas, even under strong tidal motions. Through a comparison with the highly diffusive QKE scheme in its raw formulation and with simulations without tides, we have highlighted the spurious numerical effect
that tides can have, while stressing their physical importance in transforming the water masses. Nevertheless, some questions remain open.

First of all, the choice of the filter in Eq.12 is still weakly constrained. For simplicity and to stick to the original formulation proposed in Juricke et al. (2020a), we chose a simple three-points average, and achieved satisfying results. We can however
wonder if another filter could be better suited. Basically, we would like the filter to have its cutoff wavenumber $\theta_c$ to be equal to the lowest wavenumber that the advective part $\mathcal{A}$ of the scheme can effectively resolve, in order to ensure that all the noise is effectively dissipated while the well represented scales are protected from spurious damping. It is debatable whether such a clear separation between noise and physics can be drawn, as it will always rely on the definition of an arbitrary condition (such as in Kent et al. (2014) or Winther et al. (2007)). Nonetheless, following computations made by Winther et al. (2007),
the minimum wavelength accurately represented by a fourth-order central differencing scheme is approximately $N = 4.4$ grid points. Defining the cutoff wavenumber of a filter in a standard way as the wavenumber $\theta_c$ at which $\hat{\phi}(\theta_c) \approx 1/\sqrt{2}$ (as is commonly done in signal processing), we have, for $\phi_3$, a cutoff wavelength $N_c = 2\pi/\theta_c \approx 5.5$ grid points. That is, the cutoff

of our filter is slightly greater than the actual accuracy of the advective part, ensuring that all the noisy wavelengths are effectively filtered out. On the contrary, for a higher order filter of the form

$$\phi_{5*}[s]_j = s_i + \frac{1}{16}(-s_{j+2} + 4s_{j+1} - 6s_j + 4s_{j-1} - s_{j-2}) \tag{18}$$

which transfer function writes

$$\hat{\phi}_{5*}[k] = 1 - \frac{1}{4}(1 - \cos\theta)^2 \tag{19}$$

the cutoff wavelength is approximately $N_c = 3.8$ grid points. This is slightly lower than, but comparable to the above defined accuracy of 4.4 grid points. Considering the rather approximative aspect of these computations, we could thus still consider this other filter valid. However, the stencil required to compute such a filter is wider than for $\phi_3$ (five grid points for $\phi_{5*}$ vs. three for $\phi_3$), increasing the computational cost and numerical complexity, especially when computing boundary conditions. The $\phi_3$ filter is therefore a rather good compromise, even though other filters have not been tested in the scope of this study; but we cannot exclude that other, possibly non-linear, filters might be better suited in other situations.

Second, the choice of this filter leads to an interesting result that raises the question of choosing the UP3-F vs. UP5 scheme. Looking at Eq.15, it turns out that except for the multiplying coefficient $1/6$, the damping coefficient for the filtered formulation of the UP3 scheme $\gamma^{\text{UP3-F}}$ is equal to the damping coefficient for a fifth-order upwind biased scheme (UP5), namely :

$$\gamma^{\text{UP5}} = \frac{8W}{60\Delta z}(1 - \cos\theta)^3 \tag{20}$$

See for instance Soufflet et al. (2016) for the derivation of $\gamma^{\text{UP5}}$. This is also illustrated in Fig.2. Basically, this means that the diffusive part of our filtered formulation of the UP3 scheme is equivalent to the diffusive part of a UP5 scheme, that is, a trilaplacian operator. This result is similar to the one described in Juricke et al. (2020a), where the filtered harmonic diffusion boils down to a biharmonic diffusion when $\alpha$ is set to 1. We can therefore wonder if it would not be preferable to use directly a UP5 scheme instead of a filtered scheme, since it should achieve overall a higher order of accuracy. The answer is : most likely. There are however several aspects to be considered here. Though it might be negligible in comparison to the cost of computing other components of the model, the computational cost of the UP5 scheme is slightly higher than the one of the UP3-F formulation, since in UP3-F, only the dissipative component uses a larger stencil. But more importantly, the base scheme actually used in Symphonie is QKE. It should be possible to build a scheme similar to QKE on the UP5, but some additional work would be necessary and has not been carried out in the scope of this study. Moreover, we should keep in mind that Eq.15 is derived from Eq.14 in the particular case where $\alpha = 1$. Though other values of $\alpha$ have not been discussed here, formulations where $\alpha \neq 1$ (and especially $\alpha > 1$) are believed to have some potential in further reducing numerical mixing, and will be investigated in further studies. Taking a step back, this result nevertheless suggests that a bilaplacian diffusion is still too dissipative, while a trilaplacian operator could achieve better results, at least in the situation studied here. This result might be of importance for existing and upcoming numerical cores, since increasing the order of the diffusion might be a rather affordable option.

## 5 Conclusions

In this paper, we have presented a new way of formulating the vertical advection scheme in the Symphonie model that builds on a previously available scheme and aims at making its diffusive component more scale-selective, thus reducing spurious numerical mixing, especially in a context where tides are explicitly resolved. This method has then been validated in a regional model of the South-East Asian Seas, known to be the generation site of strong internal tides that dissipate in the semi-enclosed basins of the region. By running simulations with and without tides and with the new formulation turned on and off, we have shown that the numerical diffusion is reduced enough so that the model is able to satisfyingly represent observed water masses throughout the various seas and surface fields. At the same time, we provided a clear illustration of the spurious effect that tides can have in a fixed coordinates model, even in a regional model forced at its lateral boundaries. The impact of spurious mixing on the SST is particularly important, considering its non-linear interactions with the atmosphere, as illustrated by the large fresh bias at the surface observed in the highly diffusive simulation. The effect can be even worse in fully coupled atmosphere-ocean simulations, as the whole atmosphere will adjust to the modified SST, potentially biasing the entire response of the model.

This issue of spurious numerical mixing induced by tides, although previously known, has, to our knowledge, not been as explicitly reported before as in this study. We believe this to be helpful for the growing community of model users not necessarily well-versed in numerical modelling. We advocate for a better recognition of spurious vertical mixing and its effect, especially in the context of long simulations run with explicit tidal forcing, and of the importance of choosing carefully the proper numerical methods.

Some aspects however still remain uncertain and call for further work. spurious effect of numerical mixing on tracer fields is rather coarse and qualitative. A more thorough quantification might be however of valuable insight to further disentangle the numerical from the physical mixing, and assess the relative contribution of each processes in the observed results, from surface fields to tracer profiles. The choice of such a method is however not straightforward, each exhibiting their own strengths and limitations (see e.g. Banerjee et al., 2023).

Focusing more specifically on the advection scheme presented in this paper, the choice of the filter to be used is still weakly constrained, thus calling for further investigations. A comparison to other standard schemes, though out of scope of this paper, could also be carried out. At the same time, owing to the variety of approaches in ocean modelling, defining a universal advection scheme is still difficult, if not delusional. The choice of a method indeed always involves a compromise between the quality of the solution - which is by itself oddly defined and depends on the problem itself - algorithmic complexity - which might prevent the use of a particular method on a particular set of coordinates for example - and computational cost. This is also the reason why modern dynamical cores usually offer users the possibility to choose between several advection schemes, depending on their needs. As stated in Gerdes et al. (1991), as long as the theoretical basis of a scheme is justified, the realism of the resulting simulation when compared to observations is the only way to define what's "best". The method presented here should therefore be seen more as a new element in a much wider toolbox of solutions aiming at reducing numerical errors in ocean models rather than a definitive solution to the issue of spurious vertical mixing.

*Code and data availability.* The code to plot the figures can be downloaded on the following github repository : https://github.com/Clapinet/SpuriousMixingSEA. Preprocessed Symphonie and ARGO data as well as the source code of the model (v312) are accessible via the following url https://doi.org/10.5281/zenodo.10715502. Considering their size, raw 3D model outputs are not stored publicly, but are available upon request. More information on the Symphonie model can be found on the SIROCCO gorup website : https://sirocco.obs-mip.fr/ .GLORYS data are accessible via https://doi.org/10.48670/moi-00021. OSTIA data are accessible via https://doi.org/10.48670/moi-00168. ARGO data have been downloaded from https://dataselection.euro-argo.eu/.

## Appendix A: Location and amount of ARGO floats in the region

Figure A1 shows the distribution of ARGO profiles in the region. In the interior basins, all data iver the period of interest have been selected. In the Eastern Indian Ocean, considering the large amount of data available, only a subset of profiles sampling the exit of the ITF has been selected.

## Appendix B: Metric for profiles comparison

For each zone $b$ and each depth level $k$, the standard deviation $\sigma_k^b$ of the ARGO data in the zone is computed. The RMSE is then computed by weighting each depth level $k$ by $w_k^b = (1/\sigma_k^b)/(\sum_j 1/\sigma_j^b)$. This modified metric is referred to as *weighted* RMSE (wRMSE), and writes as follow :

$$\text{wRMSE}[X_b^s, X_b^r] = \sqrt{\sum_k w_k^b (X_b^s[k] - X_b^r[k])^2} \tag{B1}$$

where $\{X_b^s[k]\}_k$ is a profile sampled at $m$ depth levels $\{k\}$ in zone $b$ for a given simulation $s$ and $\{X_b^r[k]\}_k$ a reference profile (e.g. ARGO). Choosing weights as such means that less importance, compared to classical RMSE, is given to depth levels where the *observed* variations quantified by $\sigma_k$ are greater; and conversely more importance is given to levels with less variation between the various profiles.

*Author contributions.* AG, PM and MH contributed to the design of the study. MH, JP and PM designed the configuration. MH ran the simulations. PM implemented the methods. AG analysed the simulations, carried out the theoretical work and wrote the manuscript with contributions from all coauthors.

*Competing interests.* The authors declare no competing interests.

*Disclaimer.*

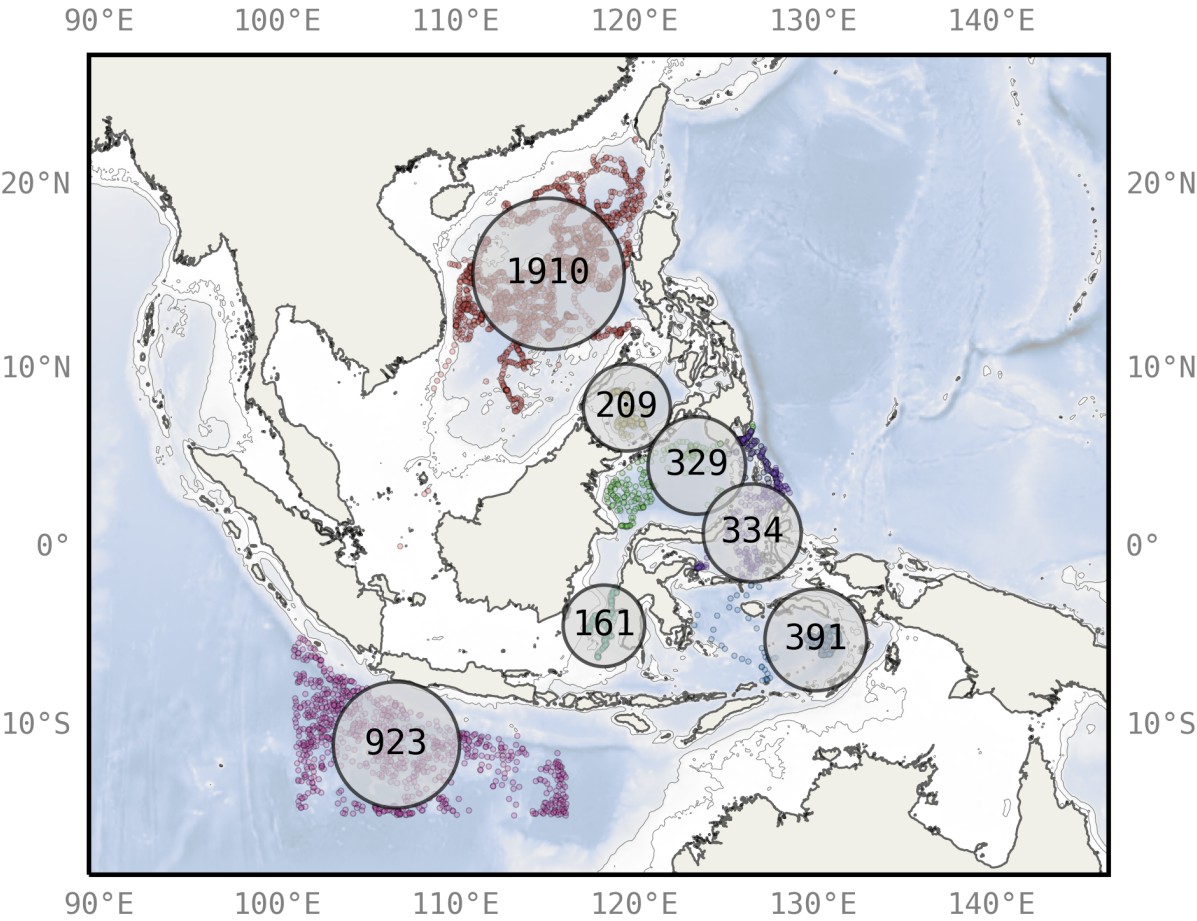

**Figure A1.** Locations and number of ARGO floats in each basin of the region over the 2017-2018 period. Individual profiles are represented by colored points, while the numbers in larger circles indicate the number of profiles in each basin over the period of interest. Basin names are provided in Fig.1.

*Acknowledgements.* The authors would like to thank the team of the Calmip computer for their support (project p20055). Analysis of simulations output was carried out using `python` the `xarray` library (Hoyer and Hamman, 2017). Heavy use of the `cmocean` package (https://matplotlib.org/cmocean/) was also made; and we would like to cite Rougier (2021) as a source of inspiration. AG finally thanks Franck Dumas, Yves Morel and Rosemary Morrow for the fruitful discussions.

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
