# Peer review of "Spurious numerical mixing under strong tidal forcing: a case study in the South East Asian Seas using the Symphonic model (v3.1.2)"

_EGUsphere, 2024_

## Referee Comment (RC2)

**1 General comments**

The manuscript is devoted to the study of tidally-driven numerical mixing in the $z$-layer SYMPHONIE ocean model. Spurious mixing is not computed as a diagnostic quantity. A rather practical approach is followed which consists in an extensive validation of the tidal and non-tidal runs against several temperature and salinity datasets. The comparison shows that, for a third order upwind advection scheme, the non-tidal simulation reproduces unrealistic sharper profiles while the tidal simulation is overly diffusive. An filtered advection scheme is introduced to cure the problem and reduce spurious mixing in tidal simulation. The manuscript is interesting and clear in both the analysis and the numerical experiments. It shows it is possible to perform tidal and internal tide simulations with a $z-$layer model, at the price of using very high order advection schemes. It shows also how different processes may balance in ocean models in an unpredictable way (in non-tidal simulation, spurious mixing compensates for the absent tidally-driven mixing and the more diffusive scheme outperforms the filtered one). The comparison with the state-of-the-art approaches that solve the problem of tidally-driven spurious mixing in other ways (smarter choice of the vertical coordinate, for example) could have been pushed further. As an aside, in some secondary discussions, I have found the style a little bit intricated.

**2 Specific comments**

1/ Line 6. Unclear sentence "This papers provides a clear illustration of this phenomenon in the context of simulation of the South East Asian Seas is provided"

2/ Line 12. I found the sentence "Simultaneously, an improvement of this advection scheme to make it more suitable for use on the vertical is provided" a little bit twisted style.

3/ Line 20. Is it more correct to talk about "diffusion" instead of "dissipation"?

4/ Line 53. "Considering the eulerian nature of the problem", what problem? Maybe rephrase: "Considering the Eulerian nature of geopotential coordinates".

5/ Line 53. I could agree with you that the adaptive or the $z-$tilde coordinates remain a research topic and their implementation into already existing numerical codes requires considerable effort. But, if the aim is to have accurate internal tide simulations, I would mention the $z$-star, which belongs to Arbitrary-Lagrangian-Eulerian coordinates. It has been around for over 20 years, easy to code in $z$-layers models and it is implemented in many operational models nowadays. Since the seminal work of [Adcroft and Camping, 2003, Ocean Modelling] it has demonstrated to outperform fixed $z-$ layers for internal tide simulations, reducing tidally induced numerical mixing. I know that it is beyond the scope of the paper but it would be very interesting to add a 5th run with the current advection scheme (UP3) and $z-$star. It is mandatory to comment on $z-$star in the introduction, it is not mandatory to add the

comparison.

6/ Line 211. Symbol $R$ (real part?) is undefined.

7/ Line 230. "and finally its limited stencil size does not increase the computational cost of the scheme too much". I kindly suggest to make this sentence more precise by computing the number of points in the stencil for both the scheme that update the tracer with UP3 (7 points?) and with the filtered version (9 points?).

8/ Which is the order of accuracy, order of truncation error of the filtered scheme? It would be very illustrative to report the curves of the damping coefficient in term of $\theta$ for UP3, the filtered scheme and a fifth-order finite difference scheme.

9/ Does the filtered scheme run stable with the same timestep of the UP3?

10/ Line 109. Could you please add the range of the vertical resolution (e.g "60 levels with varying resolution $1\,m - 100\,m$"). Spurious mixing, as shown in your analysis is strongly influenced by $\Delta z$ and the reader may be able to check the values you used.

11/ Line 131. "even though horizontal advection will also be discussed". From this sentence it seems that horizontal advection is included into the analysis but it is not. Please remove or move this sentence.

12/ Line 168. "Such non-linear schemes are however computationally costly and still exhibit a sensible amount of spurious numerical diffusion, comparable to the physical one (see e.g. Megann, 2018), with the additional drawback that it cannot be explicitly diagnosed." I find this sentence quite unclear. The first part is at least debatable. Why should vertical TVD schemes be costly (if only the horizontal grid is partitioned there is no issue in terms of parallelization and i don't think we have to worry about a few more operation to compute the slope limiter in a complex ocean model and on modern HPC architecture). Sorry I have not read your reference but with respect to what would the slope or flux limiters increase numerical diffusion? The TVD scheme has to be compared with your (4) which uses a blending between a low-order scheme and a high-order one. Why should your blended scheme (4) outperform TVD schemes? Finally why numerical mixing cannot be diagnosed? I would remove the whole sentence which is not directly related to the manuscript topic.

13/ Section 2.3.2. For sake of clarity could you please add the explicit formula for the UP3 flux, something like:

$$F_{j+1/2}^{UP3} = W_{j+1/2}\frac{T_{j+1} + T_j}{2} + ... \tag{1}$$

Then it can be easily coded by someone else.

14/ I am curious about the physical (not numerical) mixing induced by the tide in you model. This seems to be crucial in the correct reproduction of water masses, in fact, the non-tidal simulations cannot reproduce this mixing and compute very sharp tracer profiles. But I believe the hydrostatic models cannot compute explicitly this physical phenomenon. How it is recovered? Could you comment more on this in the introduction?

Thank you very much

---

## Author Comment (AC1)

We would like to begin by sincerely thanking the reviewesr for dedicating their time and attention to our paper, as well as for the positive and constructive feedback. We have carefully considered and addressed all comments and suggestions in the revised version of our manuscript.

In the attached revised version of the paper, text additions are highlighted in blue, and text removals are highlighted in red. References to line and pages numbers in the following text correspond to this highlighted version of the paper.

**RC1**

The suggestions have been taken into account throughout the text. Namely :

- a table summarising the experiments has been added (Table 1., p. 15);
- more details on the derivation of Eq. (8) have been added (lines 217 to 230 mostly, with some additions in Sec. 2.3.1, lines 163-164 to clarify some notations);
- line 235, the indices have been corrected (i -> j) and are now consistent with the rest;
- line 554 : the line has been removed;
- finally, the various typos spotted have been corrected.

We develop below on the following comment :

- *Lines 332-333. I was wondering why the T1 does not lead to any improvement to the T0 in the salinity near the surface. I suggest maybe you can briefly discuss why this is the case.*

**On the impact of tidal forcing on sea surface salinity**

- This comments seem to originate from the observation that in Fig.[4] displaying comparison of mean ARGO profiles to simulations output in the Molucca Sea, tidal simulations (T0 and T1) seem to exhibit a positive salinity bias compared to observations. This bias is notably less sensitive in simulations without tides (NT0 and NT1), which follow observations more closely at the surface.
- Looking at the dispersion of the values for the first interpolated level (5m) in both ARGO and Symphonie data throughout the various simulations, it seems that this pattern is not systematic. For instance, in the Makassar Strait or the Banda Sea there is little difference between tidal and non-tidal simulations; while in the Sulu Sea tidal simulations are actually *closer* to ARGO data than non-tidal ones.
- Now, the exact reason why such differences are observed in the Molucca Sea are unclear, but it is likely to be the result of a spurious statistical effect. Indeed, the ARGO floats do not sample the zone uniformly over the simulation period, especially when the floats present in the basin are advected in and out by the larger scales current, and so might spend more or less time sampling water masses there.
  While this effect is not an issue at depth since the variations are weaker than at the surface, or for more enclosed basins such as the Banda and Sulu Seas, the Molucca Sea is a transition region between the Pacific and the Celebes Sea, and the variablity of the complex currents system in the neighbouring Pacific ocean influences directly the amount of profiles in the basin.
- As the bias is not systematic, we haven't however investigated more this specific issue.
- Nevertheless, in order to get a more synoptic view of the effect of tides on the SSS, comparisons to gridded satellite products are also carried out. Such comparisons had been dismissed so far because of the high uncertainties of SSS measurements from space, especially in coastal areas (which make out a large part of the Indonesian Seas) associated with uncertainties on the rivers forcing used

(GLOFAS), which seem to overestimate freshwater discharges from rivers. Still, large scales comparisons of simulations outputs to SMAP-SMOS composite exhibit an interesting pattern : while NT0 and NT1 show little differences, T0 and T1 exhibit huge differences in terms of SSS, with a large freshwater bias spanning the entire Indonesian Sea in T0 while the SSS in T1 seems to be overall slightly overestimated.

- We have therefore added a new section to the revised paper (Sec. 3.4), including a discussion on this topic.
* * ** * ** * *
**RC2**

**Specific comments**

**1/ Line 6. Unclear sentence ”This papers provides a clear illustration of this phenomenon in the context of simulation of the South East Asian Seas is provided”**

As pointed out by the first reviewer, the word "provide" appears twice in this sentence. We removed the second occurrence (line 7), which should make the meaning clearer.

**2/ Line 12. I found the sentence ”Simultaneously, an improvement of this advection scheme to make it more suitable for use on the vertical is provided” a little bit twisted style.**

Indeed, we slightly change the style to make it more straightforward (line 12).

**3/ Line 20. Is it more correct to talk about ”diffusion” instead of ”dissipation”?**

In the context of tracers such as heat or salt, it is indeed probably better to talk about *diffusion*. Dissipation applies more for energy. We have modified line 20.
* * *
**4/ Line 53. ”Considering the eulerian nature of the problem”, what problem? Maybe rephrase: ”Considering the Eulerian nature of geopotential coordinates”.**

The sentence has been corrected to be more precise (lines 57-59). See also response to next comment.
* * *
**5/ Line 53. I could agree with you that the adaptive or the z−tilde coordinates remain a research topic and their implementation into already existing numerical codes requires considerable effort. But, if the aim is to have accurate internal tide simulations, I would mention the z-star, which belongs to Arbitrary-Lagrangian-Eulerian coordinates. It has been around for over 20 years, easy to code in z-layers models and it is implemented in many operational models nowadays. Since the seminal work of [Adcroft and Camping, 2003, Ocean Modelling] it has demonstrated to outperform fixed z− layers for internal tide simulations, reducing tidally induced numerical mixing. I know that it is beyond the scope of the paper but it would be very**

**interesting to add a 5th run with the current advection scheme (UP3) and z−star. It is mandatory to comment on z−star in the introduction, it is not mandatory to add the comparison.**

This comment points to a lack of clarity on our side. Lines 53 (in the first version of the paper) refers to adaptive / $z\sim$ coordinates.

The model we are using indeed also makes use of a "Quasi-Eulerian" approach (what would be referred to as "z*" in a geopotential context, though here our model is terrain following), which are natural in terrain-following coordinates with a free surface.

Our initial formulation is confusing. We improved it by adding a small discussion on z* (lines 27-32) and clarifying the sentence mentioning Lagrangian coordinates (lines 57-59).
* * *
**6/ Line 211. Symbol R (real part?) is undefined.**

Indeed, we corrected this when taking into account comments by the first reviewer (lines 163-164). Overall, we added some content in Section 2 to clarify the theoretical developments (lines 217-230).
* * *
**7/ Line 230. "and finally its limited stencil size does not increase the computational cost of the scheme too much". I kindly suggest to make this sentence more precise by computing the number of points in the stencil for both the scheme that update the tracer with UP3 (7 points?) and with the filtered version (9 points?).**

We updated the text, taking your comment into account (line 273).
* * *
**8/ Which is the order of accuracy, order of truncation error of the filtered scheme? It would be very illustrative to report the curves of the damping coefficient in term of θ for UP3, the filtered scheme and a fifth-order finite difference scheme.**

Since the Taylor expansion of the scheme is the addition of the Taylor expansion of the "dispersive" (order 4) and the "diffusive" parts (moved from order 3 to order 5), the order of accuracy of the resulting scheme is 4, at least what the tracer gradient is concerned.

Comparison of the damping coefficients for the various schemes are now provided in Fig. 2 (p.11).
* * *
**9/ Does the filtered scheme run stable with the same timestep of the UP3?**

If only the UP3 component is considered, we can show using the formulae provided in [Lemarie et al. (2015), Ocean modelling] that the UP3-F is (very slightly) more stable than UP3, with a CFL condition around 0.507 instead of 0.471.

However, the upper bound of stability for the full scheme (including the UP1 part) is set by the weighting between UP3 and UP1 combined with the fact that evaluation of the UP1 term is lagged in time. The overall stability of the scheme therefore stays limited by the forward integration of the UP1 with time step $2\Delta t$, that is $W\Delta t/\Delta z \leq 0.5$.

We have added a comment at the very end of Sec.2.3.4.
* * *
**10/ Line 109. Could you please add the range of the vertical resolution (e.g "60 levels with varying resolution 1 m − 100 m"). Spurious mixing, as shown in your analysis is strongly influenced by Δz and the reader may be able to check the values you used.**

Since the coordinates used are "quasi-sigma", this resolution varies over the domain, and ranges from around 2 m at the surface to less than 600 m for the last level at the deepest locations.

A comment on this matter has been added (line 135).
* * *
**11/ Line 131. "even though horizontal advection will also be discussed". From this sentence it seems that horizontal advection is included into the analysis but it is not. Please remove or move this sentence.**

The sentence has been removed (line 162).
* * *
**12/ Line 168. "Such non-linear schemes are however computationally costly and still exhibit a sensible amount of spurious numerical diffusion, comparable to the physical one (see e.g. Megann, 2018), with the additional drawback that it cannot be explicitly diagnosed." I find this sentence quite unclear. The first part is at least debatable. Why should vertical TVD schemes be costly (if only the horizontal grid is partitioned there is no issue in terms of parallelization and i don't think we have to worry about a few more operation to compute the slope limiter in a complex ocean model and on modern HPC architecture). [...] with respect to what would the slope or flux limiters increase numerical diffusion? The TVD scheme has to be compared with your (4) which uses a blending between a low-order scheme and a high-order one. Why should your blended scheme (4) outperform TVD schemes? Finally why numerical mixing cannot be diagnosed? I would remove the whole sentence which is not directly related to the manuscript topic.**

Indeed, TVD schemes are not necessarily more costly - a confusion has been made with schemes of the FCT family, such as the one used in NEMO. The corresponding sentence has been updated (line 200).

Concerning the second point (additional numerical diffusion), TVD schemes create numerical mixing through the "activation" of the lower order (UP1) scheme in order to preserve monotonicity in situation which would otherwise create an overshoot : since they are usually used in combination with purely dispersive schemes (e.g. second order centered schemes), they increase numerical diffusion in the sense that they actually are the source of it.
Now, we can wonder what would be the effect of adding a TVD-like formulation on top of an already diffusive scheme, such as the one proposed here : we could indeed expect the slope limiter to be activated less than when the high order scheme is purely dispersive, and thus the overall numerical mixing would be dominated by the one of the higher order scheme. This is however not directly related to this study.

Concerning the third point (blended scheme and performance), we would like to emphasise first the fact that the blending with the low-order scheme is driven by a matter of numerical stability and accuracy at the few locations of higher Courant numbers (see answer to question 9).

Secondly, our resulting scheme is not meant to "outperform" TVD scheme, for several reasons :

1. evaluating performance relies on the choice of metric, and defining a metric for a "good" advection scheme is a topic by itself;

2. even assuming the performance metric to be related to the amount of numerical diffusion associated with the scheme (which would make sense in our context), we expect routinely used TVD schemes to produce at least as much numerical mixing as our filtered version (and probably more): see for instance comparisons carried out for MITgcm here -> https://mitgcm.readthedocs.io/en/latest/algorithm/adv-schemes.html#comparison-of-advection-schemes. However, a direct analytical comparison is not possible (see paragraph below);

3. last, and maybe more importantly, the formulation used in our study allows us, by simply switching the filtering on or off (or more generally play on the coefficient $\alpha$, as mentioned in section 2.3.4, equation 14 of the updated version), to act only on the "amount" of numerical diffusion introduced in the model. In this sense, it is a useful tool to illustrate the effect of numerical mixing.

Finally, for such schemes making use of non-linear limiters, numerical mixing cannot be (to the best of our knowledge) analytically diagnosed, at least not in the sense considered here : that is, either by extracting an explicit diffusion operator from a Taylor expansion or by computing a spectral damping coefficient. This is the reason why several experimental diagnostics methods have been proposed in the literature (see section 3.1), since a number of schemes make use of non-linear limiting procedures.

We updated the sentence by removing the part concerning the computational cost (line 200). We however kept (but slightly reformulated) the part on the possible drawbacks of non-linear schemes (line 202), as we believe that it is important to mention other kind of approaches used to prevent dispersion errors from growing - even though it is not the purpose of the paper. Nonetheless, if you believe that this part is harmful to the paper, we can remove it.

**13/ Section 2.3.2. For sake of clarity could you please add the explicit formula for the UP3 flux, something like [...]**

In the revised theoretical part, the formulation of the dissipative part is given, and we point out to two different papers for the full details of the scheme, so as not to add too many equations not directly necessary to the discussion in the text.

**14/ I am curious about the physical (not numerical) mixing induced by the tide in you model. This seems to be crucial in the correct reproduction of water masses, in fact, the non-tidal simulations cannot reproduce this mixing and compute very sharp tracer profiles. But I believe the hydrostatic models cannot compute explicitly this physical phenomenon. How it is recovered? Could you comment more on this in the introduction?**

Note first that no specific parameterisation has been added in the tidal simulations. The vertical diffusivity characterising physical mixing is computed solely via the turbulent closure equation, i.e. $k - \epsilon$ here. Difference in the amount of physical mixing in tidal and non-tidal simulations is therefore only related to the way the (internal) tide dynamics will impact the various components of the closure equations - namely via the vertical shear and buoyancy terms.

Whether such parameterisations are adapted to tidal simulations is still unclear. Many studies seem to give

good results, but it is unclear whether it is "out of luck" or because they can indeed capture the leading drivers of mixing.

We have updated Section 2.1 with a discussion on the origin of mixing in hydrostatic models (lines 100-116); and we provide below some additional considerations, which we do not believe to be of direct interest for the paper, but that might interest you nonetheless.

Here are a few comparison of $K_z$ as computed by the model between a tidal (T1) and a non-tidal (NT1) simulation.

[Figure]

This represents the median values of $K_z$ (in $\log m^2.s^{-1}$) over one month (2018/01) of simulation, along a slice at 129°E longitude, cutting through the Halmahera, Ceram and Banda Seas and known for being a hotspot of internal tides generation. Median values were used instead of mean, as transient phenomenon can increase locally $K_z$ by a few order of magnitudes, thus biasing the mean and making the picture "noisier". Note however that the median has been computed with already daily averaged values.

The same analysis is computed at another region known for being the generation zone of powerful internal waves, the Luzon Strait (taken at 21°N), with a slice extending up to the continental shelf :

[Figure]

Looking at those comparisons, we can see that in T1, vertical diffusivity is enhanced over rough topography, mostly close to the bottom boundary but sometimes extending up to a few hundreds of meter above it. It is likely this is the result of the interaction of tidal currents with the seafloor plays an important role, though daily averaged output do not really allow to diagnostics their contribution. The shear at the bottom boundary will then "propagate" vertically, leading to the observed vertical structures.
Some other local phenomenon linked directly to internal waves generation and dissipation at straits might also be (partially?) resolved, though we must concede a lack of more knowledge on the topic at the moment.

At any rate, we are currently investigating the origine and importance of physical mixing in our model, and the results are likely to be the basis of future publications. In a similar vein, another interesting question is : is this increased physical mixing sufficient to actually reproduce the water mass transformation occurring in the Indonesian Seas, or is the remaining numerical mixing also playing an important role ?

As a concluding remark, let us emphasise that the mechanisms leading to mixing in the Indonesian Seas are still an active topic of research. It is now widely accepted that tides play a crucial role in this mixing and there are both modelling and observational evidence of hotspots of increased turbulent dissipation; but the questions of how and when this mixing exactly occurs remain rather uncertain, especially in models.

---

## Referee Report (RR1)

The main remark about the comparison with state of the art vertical coordinates (point #5) has been cleared out, the default run is already using Quasi-Eulerian vertical coordinates, as I have asked. All the other suggestions have been taken into account. I thank the authors for the interesting discussion on the physical mixing (#14). I have only some minor doubts, that can be further clarified.

**1  Specific comments**

1/ Section 2.3.3. Since the default scheme is already ALE or "Quasi-Eulerian", the velocity $W$, should not be the vertical velocity at the interface between $j$ and $j+1$, but the relative velocity. I guess when you talk about "typical numerical values $W$ ..." you are already considering the "good" velocity. If this is the case, using another name for $W/\tilde{w}$ instead of the generic "velocity field" could help to further clarify my original misunderstanding.

2/ The paragraph about TVD is a slightly more clear to me now. Although I still do not fully understand the comparison between a high order linear scheme such as UP3 and non-linear ones such as TVD/WENO that of course are more diffusive since they are designed to handle sharp gradients.

3/ Thank you for the discussion on the stability. Could you please clarify what do you mean when you write "Since the upper bound is set by the fact that the evaluation of the UP1 term is lagged in time."?

Thank you

---

## Author Response (AR2)

We thank the reviewers for taking the time to look at our work and for their pertinent and constructive comments, which helped to improve the quality of the paper by making the discussion clearer.

As in the first round of review, in the attached revised version of the paper, text additions are highlighted in blue, and text removals are highlighted in red. References to line and pages numbers in the following text correspond to this highlighted version of the paper.

**1/ Section 2.3.3. Since the default scheme is already ALE or "Quasi-Eulerian", the velocity W , should not be the vertical velocity at the interface between j and j + 1, but the relative velocity. I guess when you talk about "typical numerical values W ..." you are already considering the "good" velocity. If this is the case, using another name for W /$\tilde{w}$ instead of the generic "velocity field" could help to further clarify my original misunderstanding.**

Indeed, we did not update the theoretical discussion while adding the clarification on ALE coordinates. We thus added a paragraph (lines 175-180) to clarify this.

**2/ The paragraph about TVD is a slightly more clear to me now. Although I still do not fully understand the comparison between a high order linear scheme such as UP3 and non-linear ones such as TVD/WENO that of course are more diffusive since they are designed to handle sharp gradients.**

We understand your comment and removed the corresponding sentence (lines 202-204).

**3/ Thank you for the discussion on the stability. Could you please clarify what do you mean when you write "Since the upper bound is set by the fact that the evaluation of the UP1 term is lagged in time."?**

We realize our discussion on this topic was indeed rather vague. We provided more details in the text (lines 289-296).